# Seasonality and synchrony of photosynthesis in African forests inferred from spaceborne chlorophyll fluorescence and vegetation indices

Russell Doughty[1*], Michael C. Wimberly[2], Dan Wanyama[2], Helene Peiro[1,3], Nicholas Parazoo[4], Sean Crowell[5], Moses Azong Cho[6]

[1]College of Atmospheric and Geographic Sciences, University of Oklahoma, Norman, OK, 73019, USA
[2]Department of Geography and Environmental Sustainability, University of Oklahoma, Norman, OK, 73019, USA
[3]Netherlands Institute for Space Research (SRON), Leiden, The Netherlands
[4]Jet Propulsion Laboratory, California Institute of Technology, Pasadena, CA 91109, USA
[5]LumenUs Scientific Solutions, LLC, Oklahoma City, Oklahoma.
[6]Precision Agriculture Research Group, Advanced Agriculture and Food, CSIR, Pretoria, South Africa

*Correspondence to*: Russell Doughty (russell.doughty@ou.edu)

**Abstract.** Global atmospheric carbon dioxide concentrations are largely driven by changes in fossil fuel emissions and terrestrial photosynthesis, of which tropical forests account for one third. Relative to other tropical regions, less is known about the seasonality of African tropical forest productivity and its synchrony with environmental factors due to a lack of in situ carbon flux data. To help fill this knowledge gap, we use spaceborne solar-induced chlorophyll fluorescence (SIF), vegetation indices—including the Enhanced Vegetation Index (EVI), Normalized Difference Vegetation Index (NDVI), and Land Surface Water Index (LSWI)—and climate data to investigate the seasonality and synchrony of indicators of photosynthesis in Africa's tropical forest ecoregions. We find West African SIF to increase during the dry season and peak prior to precipitation, as has been observed in the Amazon. However, NDVI and EVI do not mimic the strong double-peak seasonality observed in SIF; instead, they often plateau until substantial decreases occur in the dry season. In Central Africa, we find a continental-scale bimodal seasonality in SIF and EVI, the minimum of which is synchronous with precipitation, but its maximum is likely less related to environmental drivers. Our findings highlight the complex relationships between SIF, vegetation indices, and environmental factors, underscoring the importance of using multiple remote sensing measures to monitor tropical forest productivity.

## 1 Introduction

The intra- and inter-annual variability of global atmospheric carbon dioxide concentration is driven largely by changes in the terrestrial uptake of carbon dioxide through photosynthesis, and tropical forests are playing an increasingly important role in this variability (Wang et al., 2014). Tropical forests account for about one-third of global photosynthesis, house one half of Earth's terrestrial carbon stock, and sequester about 15% of anthropogenic carbon dioxide emissions (Gaubert et al., 2019; Pan et al., 2011; Sitch et al., 2015). They also play important roles in global and regional water cycles via precipitation recycling

and cloud formation (Lawrence and Vandecar, 2015; Sheil, 2018; Spracklen et al., 2012; Worden et al., 2021). Thus, tropical forests are critical to regulating global climate.

Rainfall seasonality in the tropical regions of West and Central Africa are primarily driven by the monsoon and the Intertropical Convergence Zone (ITCZ) (Longandjo and Rouault 2024). These two large-scale atmospheric processes create seasonal variations that bring distinct wet and dry seasons. Most areas near the equator, including parts of Coastal West Africa and Central Africa, experience two distinct passes of the ITCZ each year, resulting in a bimodal rainy season (Nicholson and Grist 2023). Byrne et al. (2018) reported a narrowing and strengthening of rainfall in the ITCZ over recent decades, based on satellite observations and simulation studies. However, their study found no evidence of a shift in the ITCZ's location.

The floristic composition and distribution of the tropical forests in the Guineo-Congolian region of West and Central Africa remain poorly sampled and understood (Sosef et al., 2017). Despite this, there is a general consensus that rainfall significantly influences floristic patterns across the region (Fayolle, 2014). Various authors have classified African tropical forests in different ways over time, including White (1979). However, a recent study by Fayolle et al. (2014) categorizes these forests into four groups: Wet-moist West Africa, Dry West Africa, Wet Central Africa, and Moist Central Africa. Using the RAINBIO database of tropical African vascular plant species, Sosef et al. (2017) reported a total of 22,577 species in the region. However, the authors emphasized that tropical forest biodiversity is still inadequately sampled.

Relative to the Amazon basin of South America, less is known about the seasonality of photosynthesis of African tropical forests, their drivers, and their responses to changes in climate due to a lack of eddy covariance tower measurements in structurally intact forests (Malhi, 2012; Merbold et al., 2009; Williams et al., 2007). Responses of African tropical forest productivity to climate have instead been gleaned from syntheses of (1) field plot measurements that have focused on changes in aboveground biomass to assess gains and losses in the net carbon sink over the three decades preceding 2015; and (2) satellite remote sensing of leaf area and greenness.

Studies that have focused on field plot measurements had three main findings. First, they found a significant upward trend in carbon gains (Hubau et al., 2020) that were unaffected by anomalously low precipitation and high temperatures during the 2015/2016 El Nino (Bennett et al., 2021). Second, there was no significant trend in carbon losses, which were also not significantly affected by the 2015/2016 El Nino, despite there being a strong correlation between precipitation and the net carbon sink at the continental scale (Williams et al., 2007). Finally, there was no significant trend in the net carbon sink, but the net sink, which remained positive, was significantly reduced by the 2015/2016 El Nino. Thus, field-based evidence suggests that African tropical forests might be especially resistant and resilient to climate extremes, but additional research is needed.

Satellite remote sensing studies have identified a double peak in the seasonality of leaf area and greenness in the Congolian tropical forests, which aligns with precipitation patterns. However, little research has been published on the seasonal dynamics of West African tropical forests, highlighting a significant knowledge gap. Understanding these seasonal patterns is crucial, especially amid debates over long-term vegetation changes in African tropical forests. While some studies have suggested a significant long-term browning trend in the Congolian forests, potentially linked to large-scale drying events (Asefi-Najafabady and Saatchi, 2013; Jiang et al., 2019; Malhi and Wright, 2004; Zhou et al., 2014), the most recent research found

no widespread long-term decline in leaf area or greenness (Sun et al., 2022). This latest finding aligns with field observations
showing no significant trend in the net carbon sink, suggesting that African tropical forests may be relatively resilient to climate
variability. Addressing the lack of knowledge on the seasonality of West African tropical forests is therefore essential to fully
understand the ecological dynamics and climate resilience of these ecosystems.
Although these studies have investigated long-term changes in the net carbon sink, greenness, and leaf area, they provide little
insight into the relationship between photosynthesis and environmental factors or how photosynthesis responds to climate
anomalies. For instance, the field investigations do not provide definitive evidence for whether the decrease in the net carbon
sink during the 2015/2016 El Nino was due to decreased photosynthesis, increased respiration, or both. Also, these previous
field-based analyses aggregated measurements annually at the continental scale, although the field sampling was more
commonly conducted in coastal forests, which are located closer to the coastlines. These coastal forests have different
environmental conditions and characteristics than interior forests (Blundo et al., 2021; Lopez-Gonzalez et al., 2011), with
higher annual total rainfall and extreme variability in monthly precipitation and photosynthetically active radiation (PAR)
compared to the interior Congolian rainforest.
Recent advancements in the retrieval of solar-induced chlorophyll fluorescence (SIF) from space provides an observation-
based method for monitoring plant physiology and the amount of PAR absorbed by chlorophyll ($APAR_{chl}$) and has been
described as a proxy of photosynthesis (Doughty et al., 2019, 2021b). SIF is a small amount of energy that is re-emitted by
chlorophyll (1%-2%) and is sensitive to leaf physiology, which are the functions and processes within a plant leaf, including
how it absorbs sunlight, exchanges gases through stomata, transports water and nutrients, and carries out photosynthesis
(Johnson and Berry, 2021; Porcar-Castell et al., 2021, 2014). Because SIF is emitted during the light reactions of
photosynthesis, it is directly sensitive to both the quantity of light absorbed and the efficiency with which that light is used for
carbon fixation. This makes SIF a more direct proxy of photosynthetic activity and plant productivity compared to traditional
vegetation indices, which primarily capture canopy greenness and structure.
Thus, SIF is directly sensitive to changes in $APAR_{chl}$ and can correlate to changes in photosynthetic activity (Yang et al.,
2018), particularly at coarse spatio-temporal resolutions (Magney et al., 2020). For example, spaceborne SIF was found to
mimic the seasonality of photosynthesis estimated at eddy covariance tower sites in the tropical Amazon forest, and more
closely tracked photosynthesis than vegetation indices (Doughty et al., 2019), which have traditionally been used to estimate
$APAR_{chl}$ and to model photosynthesis globally (Pei et al., 2022).
The studies that have utilized spaceborne SIF to investigate tropical Africa have found that (1) temperature and vapor pressure
deficit (VPD)—which are interlinked because higher temperatures increase VPD by raising the air's capacity to hold
moisture—control the productivity of African tropical forests (Madani et al., 2017; Umuhoza et al., 2023); (2) SIF tracks well
the seasonality of photosynthesis, or gross primary productivity (GPP), over Africa (Mengistu et al., 2021); and (3) SIF has
weak to insignificant relationships with VIs and VI-based $APAR_{chl}$ (Doughty et al., 2021b). However, these earlier remote
sensing studies have not characterized the relationships between SIF and environmental factors for African tropical forests at
regional scales despite there being important and substantial differences in the seasonalities and variability of environmental

factors. Relatively high spatial resolution spaceborne SIF data acquired from the newest SIF platforms, including TROPOMI, OCO-2, and OCO-3, is now available and allows us to characterize the relationships between SIF and environmental factors at finer spatial scales.

Here, we leverage SIF data from these platforms to advance our knowledge on African tropical forest carbon uptake by inferring the seasonality of photosynthesis for 11 African tropical forest ecoregions from 2019 through 2021. Photosynthesis and SIF was found to be decoupled from vegetation indices and precipitation in the Amazon due to changes in leaf demography and physiology (Doughty et al., 2019; Restrepo-Coupe et al., 2013; Wu et al., 2016). An analysis of six subtropical evergreen species in Africa found the correlation between VIs and leaf nutrients, which are closely related to photosynthesis, to be seasonally dependent (Van Deventer et al., 2015). Also, studies have found tropical moist forests with more than 2000 mm mean annual precipitation (MAP) to be radiation-limited rather than water-limited (Doughty et al., 2019; Guan et al., 2015), and that photosynthesis and SIF of moist Amazon forests were positively associated with VPD (Green et al., 2020).

Thus, we suspected that leaf demography and physiology could be responding similarly in African tropical forests to changes in environmental conditions as they do in the Amazon. Our first hypothesis was that the seasonalities of SIF and vegetation indices in ecoregions with moist forest (> 2000 mm MAP) would differ substantially but would be more highly correlated in less moist forest (< 2000 mm MAP). Our second hypothesis was that SIF would be more strongly coupled with precipitation in less moist African forests and that SIF and VPD would be positively correlated in moist forest.

## 2 Methods

### 2.1 Data filtering using Copernicus Forest Cover

We used both gridded and ungridded data in our analyses and these datasets are explicitly described below. All data were filtered using the 100-m Copernicus Land Cover dataset for the year 2019 (data after 2019 is not available) (Buchhorn et al., 2020), thus we only used data that fell within the forested areas. All data was used as-is without gridding or re-gridding, and values were aggregated to monthly timesteps at the ecoregion scale. To help ensure that our spaceborne data were acquired over forest and to reduce the likelihood of mixed pixels and soundings with mixed land cover types, we converted the forest land cover raster data to polygon and created a 2.5 km inner buffer.

### 2.2 OCO-2 and OCO-3 SIF

The Orbiting Carbon Observatory-2 (OCO-2) is a NASA satellite that was launched in July 2014, and OCO-3 is a duplicate of the OCO-2 grating spectrometer that was attached to the Japanese Experimental Module Exposed Facility (JEM-EF) on the International Space Station (ISS) in May 2019 (Eldering et al., 2019). They have three bands: an oxygen-A band at 0.765 μm and carbon dioxide bands at 1.61 μm and 2.06 μm. The swath widths are ~10 km with eight measurements across-track. The spatial resolution at nadir is slightly different for OCO-2 and OCO-3, about 1.3 km by 2.25 km and 1.6 km by 2.2 km, respectively.

We used the ungridded, sounding-level data from the Level 2 v10 SIF Lite files that are available for each platform (OCO-2 Science Team et al., 2020; OCO-3 Science Team et al., 2020). We used SIF scaled to 740 nm, which is computed using retrievals from the 757 nm and 771 nm spectral windows and a reference spectral shape for SIF (Doughty et al., 2021a). Scaling to 740 nm can reduce uncertainty and allows for a better comparison among sensors as the various sensors from which we retrieve SIF have different retrieval windows. Also, we used daily adjusted values, which are scaled from instantaneous SIF values using the geometry of incoming solar radiation for that day to help account for differences in the timing of data acquisition and solar illumination angles (Frankenberg et al., 2011; Köhler et al., 2018).

## 2.3 TROPOMI SIF

The TROPOspheric Monitoring Instrument (TROPOMI) instrument is on board the Copernicus Sentinel-5 Precursor satellite, which launched in October 2017. It provides near-daily global SIF data since May 2018 at a resolution of 3.5 km by 5.5 km at nadir and has a swath width of ~2600 km. Here we used the Level 2 TROPOspheric Monitoring Instrument (TROPOMI) TROPOSIF data product (Guanter et al., 2021). We used daily averaged SIF retrievals from the 743-758 nm retrieval window as the 735-758 nm window had a higher sensitivity to atmospheric effects (Guanter et al., 2021). These data are also scaled to 740 nm values. We did not filter any of the SIF data from OCO-2, OCO-3, or TROPOMI using a cloud fraction threshold as SIF is relatively less sensitive to cloud cover than surface reflectance (Guanter et al., 2015) and we wanted to avoid a clear sky bias (Köhler et al., 2018).

## 2.4 CHIRPS Precipitation

Our precipitation data came from Climate Hazards group InfraRed Precipitation with Stations (CHIRPS), which is a long-term, near-global, daily data set. CHIRPS incorporates The Climate Hazards Group climatology (CHPclim), 0.05° resolution satellite imagery, and in-situ station data to produce a 0.05° resolution gridded data set for time series, trend, and drought monitoring (Funk et al., 2015).

## 2.5 ERA5 Reanalysis

We used monthly averaged data from the ERA5-Land product (Muñoz Sabater, 2019), which is available in a spatial resolution of 0.1 degrees, for air temperature, photosynthetically active radiation (PAR) at the top of the canopy (PAR$_{TOC}$), VPD, and volumetric soil moisture (layer 1; 0-7 cm). PAR$_{TOC}$ was calculated as a fraction (0.48) of the downward shortwave radiation from ERA5 after being converted from J/m-2 to W/m-2 by dividing the original values by the number of seconds in the month. VPD was calculated by first applying Tetens equation to air temperature ($T_{air}$) and dew point temperature ($T_{dew}$) for temperatures above 0°C (Monteith and Unsworth, 2013):

$$es = 0.61078 \times exp(\tfrac{17.267 T_{air}}{237.3 + T_{air}}) \tag{1}$$

$$ea = 0.61078 \times exp(\tfrac{17.267 T_{dew}}{237.3 + T_{dew}}) \tag{2}$$

where $es$ is the saturation vapor pressure or vapor pressure at air temperature, and $ea$ is the actual vapor pressure or vapor pressure at dew point temperature. VPD was then derived as:

$$VPD = es - ea \tag{3}$$

## 2.6 MODIS Surface Reflectance and Vegetation Indices

We used the 500-m daily MCD43A4 surface reflectance product (Schaaf and Wang, 2015) to compute four vegetation indices: the Normalized Difference Vegetation Index (NDVI), Enhanced Vegetation Index (EVI), the Near-infrared Reflectance of Vegetation (NIRv), and the Land Surface Water Index (LSWI). NDVI has been traditionally used to assess vegetation greenness (Rouse et al., 1974), but it tends to saturate in areas with a high leaf area index such as the tropics (Huete et al., 1997b). This saturation limits NDVI's ability to detect subtle changes in the forest canopies of these ecosystems.

EVI, by contrast, incorporates additional information from the blue band and accounts for atmospheric effects and canopy background signals. Thus, EVI is less prone to saturation than NDVI, particularly in regions with dense vegetation such as African tropical forests (Huete et al., 1997a). EVI is also more sensitive to variations in canopy structure and leaf area, allowing for better differentiation between areas with similar levels of greenness but different biophysical properties. Because of these advantages, EVI is a preferred metric in studies focusing on tropical forests, where vegetation indices are often challenged by the dense, multi-layered canopies typical of these ecosystems.

NIRv is a recently developed indicator that overcomes NDVI's saturation limitations by multiplying NDVI by the near infrared band, which is highly sensitive to leaf cellular structure (Badgley et al., 2017). Although NIRv shows promise for detecting vegetation dynamics, it is still relatively new and less well-validated in the context of tropical forest canopies.

LSWI is computed using the shortwave infrared band, and is primarily used for assessing leaf water content and soil moisture (Xiao et al., 2002). While LSWI offers useful insights into hydrological changes in vegetation, it is less directly related to leaf physiology and overall canopy structure. Although the focus of our manuscript is on physiology, we include LSWI to give additional insight into the seasonality of canopy water content, as water availability is important for leaf physiological processes .

The equations for these vegetation indices are as follows:

$$EVI = 2.5 \times \frac{NIR-RED}{NIR+6*RED+7.5*BLUE+1} \tag{4}$$

$$NDVI = \frac{NIR-RED}{NIR+RED} \tag{5}$$

$$NIRv = NDVI \times NIR \tag{6}$$

$$LSWI = \frac{NIR-SWIR}{NIR+SWIR} \tag{7}$$

where $NIR$ is the near infrared band, $RED$ is the red band, $BLUE$ is the blue band, and $SWIR$ is the shortwave infrared band.

**2.7 Ecoregions**

An ecoregion is a substantial geographic area characterized by a unique composition of natural communities and ecosystems, where the majority of its species, ecological interactions, and environmental conditions are distinct and unified. These regions reflect the historical distribution of specific species and ecosystems and are categorized within broader biomes like forests, grasslands, or deserts, encompassing the diversity of terrestrial life on Earth. We used the Terrestrial Ecoregions of the World boundaries (Olson et al., 2001) to distinguish between Africa's tropical forest types (Fig. 1), of which there are twelve. We combined the Nigerian Lowland Forests and the Niger Delta Swamp Forest ecoregions, which are adjacent to each other, in our analyses due to the sparsity of forest and spaceborne data for these forests.

**2.8 West African and Central African Tropical forest**

We noticed that the wettest ecoregions (Fig. 1) also had the highest variability in monthly total rainfall, and that there was a dissimilarity in our results among the wettest ecoregions with a high variability in monthly precipitation and the drier ecoregions with low variability (see precipitation bars in Figs. 2-3). Thus, we classified the 11 ecoregions into three groups according to their precipitation regime, monthly variability, and mean annual rainfall (Fig. S1, Table S1). Four ecoregions in West Africa were characterized by seasonalities in mean monthly precipitation that had distinctive single wet and dry periods each year (Figs. 2-3), high monthly variability (sd ≥ 120 mm), and relatively high mean annual rainfall (> 2400 mm). We classified these ecoregions as West African moist tropical forest, which included the Cameroonian Highlands, Cross-Sanaga-Bioko Coastal Forest, Nigerian Lowlands and Niger Delta, and Western Guinean Lowlands. The six Central African ecoregions were characterized by seasonalities that typically had a double-peak pattern, low monthly variability (sd ≤ 100 mm), and relatively lower mean annual rainfall (< 2200 mm). We classified these forests as Central African tropical forests. The precipitation regime of the Eastern Guinean ecoregion in West Africa, which we classified as West African tropical forest, had mean annual rainfall (1544 mm) and monthly rainfall variability (81 mm) that was more similar to the Central African ecoregions.

**3 Results**

**3.1 West African moist tropical forests**

**3.1.1 Seasonality of SIF, environmental factors and VIs**

We first evaluated the synchrony between SIF and precipitation for each ecoregion using lag correlations, and we found that the lag correlations were bimodal or plateaued for the West African moist tropical forests (Fig 1). SIF had a distinctive double-peak seasonality across all 11 ecoregions, but in the West African moist tropical forests the first peak in SIF was distinctively larger than the second and preceded peak precipitation (Fig. 2). SIF increased at the beginning of each year along with precipitation as temperature, PAR, and VPD decreased, but SIF peaked prior to peak precipitation and minimums in

temperature, PAR, and VPD. The second, smaller peak in SIF tended to occur as precipitation decreased and PAR increased, but before large increases in VPD. Minimum SIF coincided with minimum precipitation and peaks in temperature, PAR, and VPD. PAR exhibited a relatively strong seasonality with minimums occurring mid-year due to high cloud cover during peak precipitation.

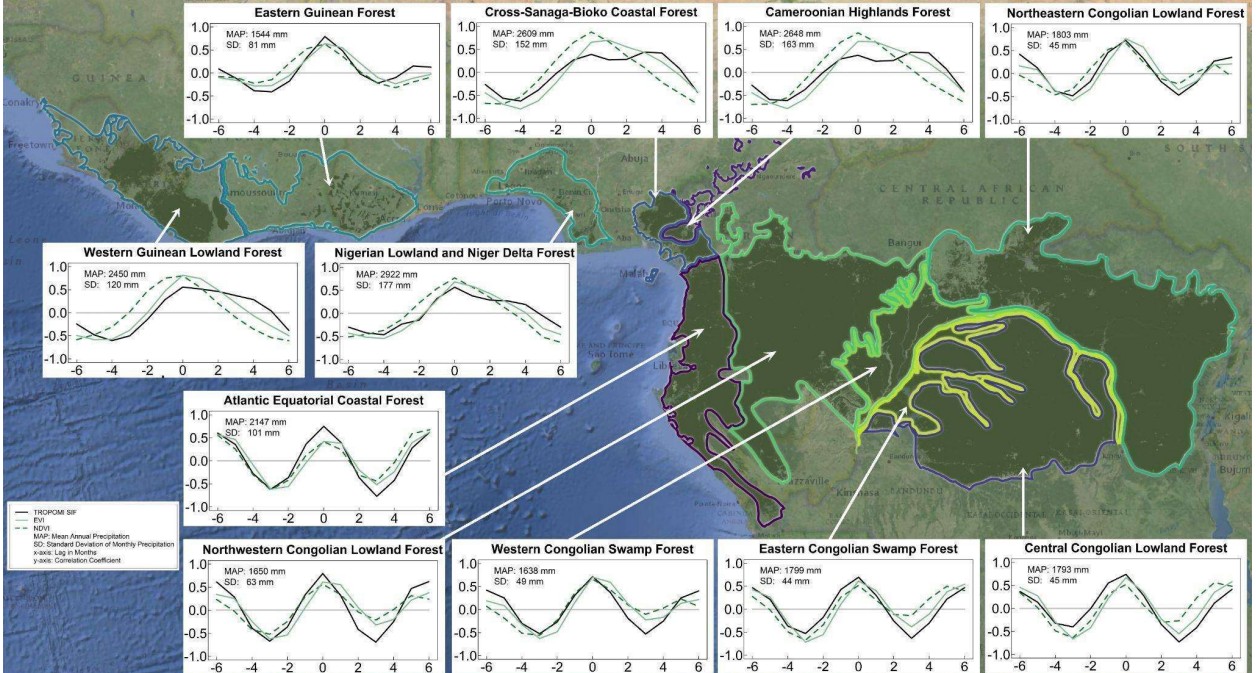

**Figure 1. Lag correlation plots between precipitation and SIF, EVI, and NDVI for 11 tropical forest ecoregions. Positive values indicate a shift of the precipitation forward in time, and negative values indicate a shift of the precipitation backward in time. Solid black line is TROPOMI SIF; Solid green line is EVI; Dashed green line is NDVI; MAP is mean annual precipitation; SD is standard deviation of monthly precipitation; x-axis is Lag in Months; and y-axis is Correlation Coefficient.**

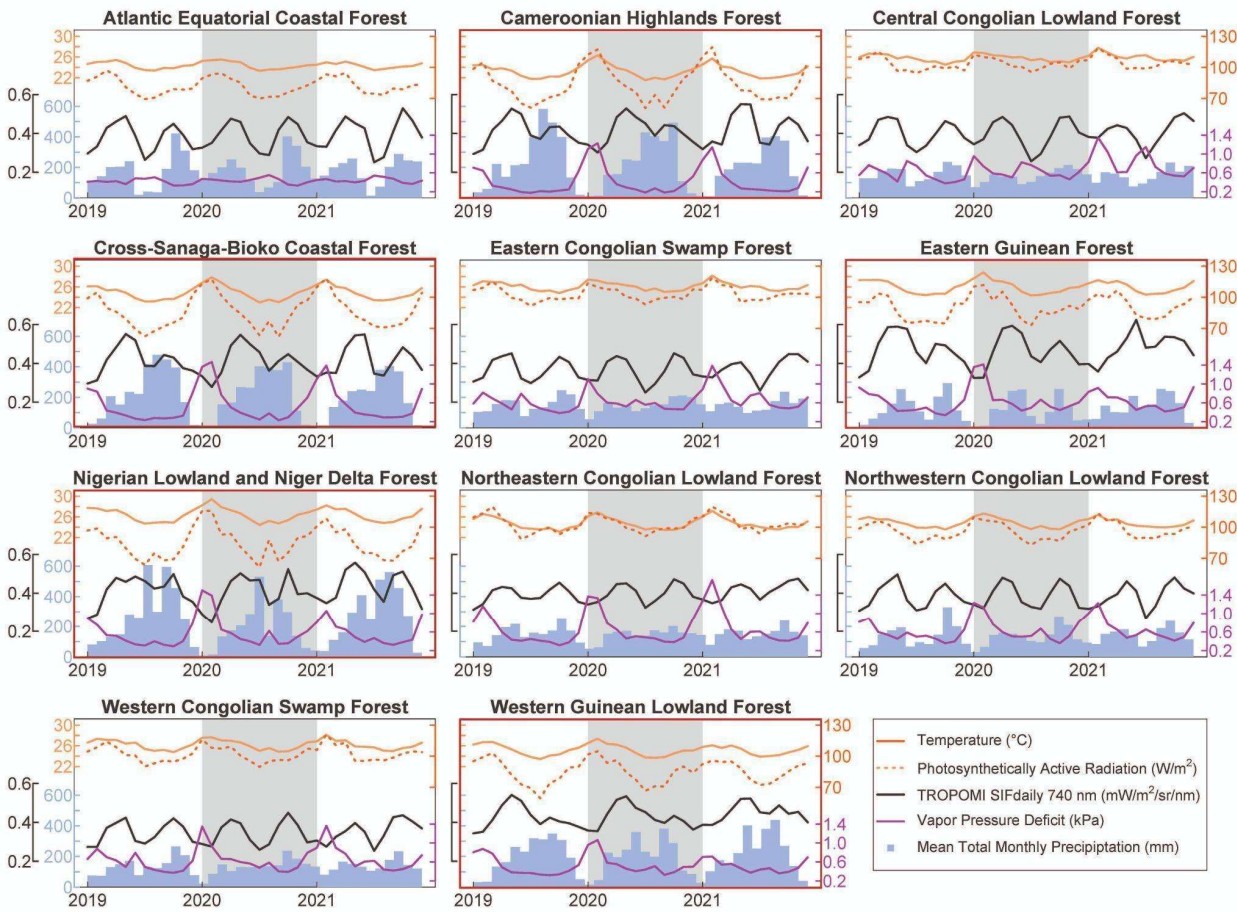

231

**Fig. 2. Environmental conditions and solar-induced chlorophyll fluorescence for 11 African tropical forest ecoregions.**
**Photosynthetically active radiation (PAR) is the amount of PAR at the top of the canopy (PAR_TOC). West African ecoregions are**
**outlined in red.**

NDVI and EVI did not mimic the double-peak seasonality we observed with SIF in the West African moist tropical forests
(Fig. 3). Although SIF, NDVI, and EVI increased at the beginning of each year as precipitation increased, NDVI and EVI
plateaued until substantial decreases in the dry season during the last quarter of the year.

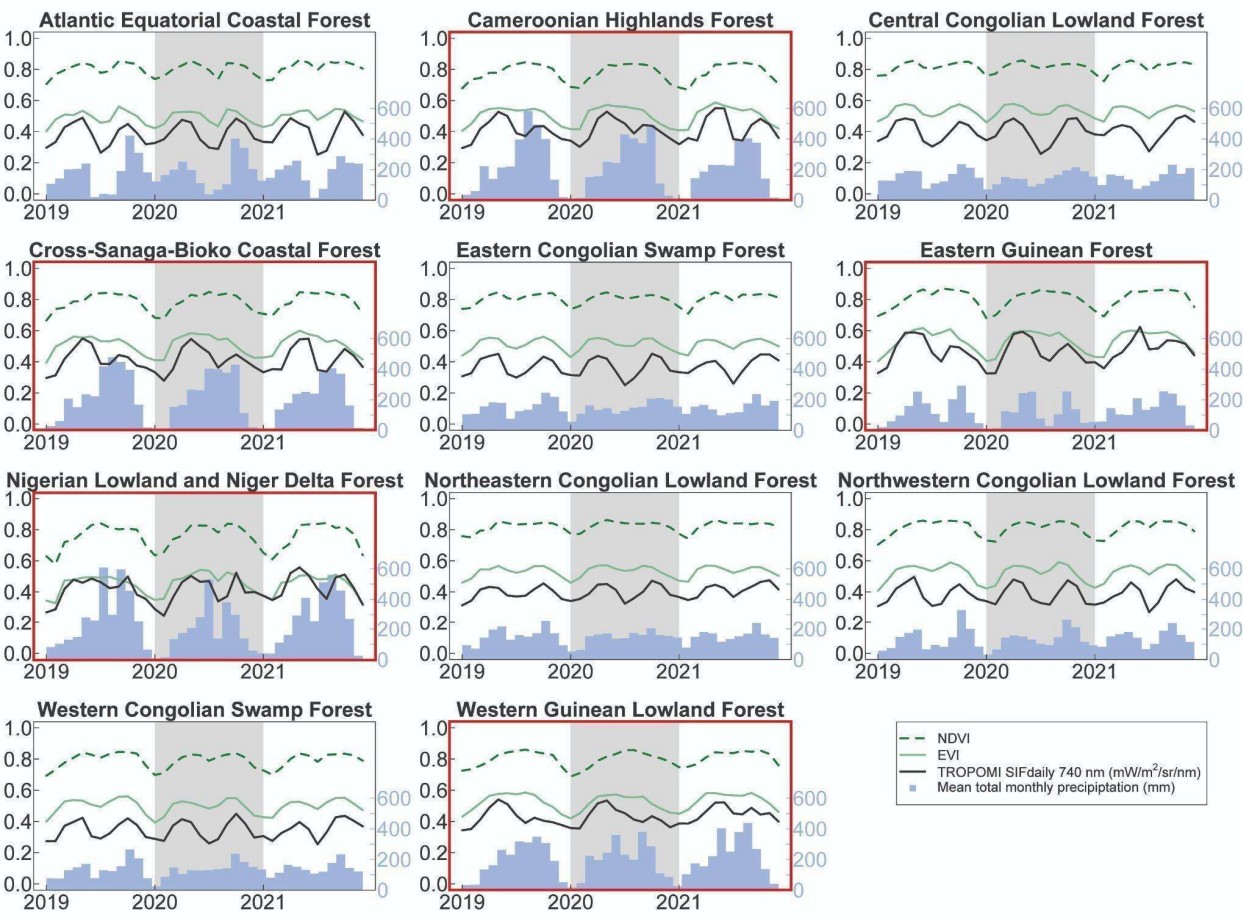

238

**Figure 3. Monthly mean NDVI, EVI, SIF, and precipitation for 11 tropical forest ecoregions of Africa for 2019 - 2021. The shaded region delineates the year 2020. NDVI, EVI, and SIF share the left y-axis. West African ecoregions are outlined in red.**

**3.2 Central African tropical forests**

**3.2.1 Seasonality of SIF, environmental factors, and VIs**

The seasonality of SIF and environmental factors in the Central African tropical forest differed remarkably from those in the West African moist tropical forest (Fig. 1). In Central Africa, the peaks and minimums in SIF tended to be similar in magnitude (Fig. 2) and were synchronous with precipitation, but there were some notable differences in the seasonalities of VPD. In the Atlantic Equatorial Coast Forest, VPD was relatively stable as the drier periods coincided with reduced temperature and PAR. In the Central Congolian Lowland Forest, VPD had a distinct double-peak seasonality that juxtaposed precipitation and SIF. In the four other Congolian ecoregions, there were large annual peaks in VPD that occurred at the beginning of the year when precipitation was low and temperatures high, but interestingly the magnitude of the decrease in SIF during this time was similar to the mid-year decrease in SIF when VPD was low.

Unlike the West African moist tropical forest, we found the seasonality of EVI to mimic SIF with a double-peak pattern in the
Central African tropical forest (Fig 3). The same was true for NDVI, except in the Northeastern Congolian Lowland where
NDVI plateaued during the wet season.

## 3.3 Relationship between SIF, environmental factors and VIs

Our second hypothesis posited that solar-induced fluorescence (SIF) would be more strongly coupled with precipitation in less
moist African forests and that SIF and vapor pressure deficit (VPD) would be positively correlated in moist forests. Our
analysis supports the first part of this hypothesis but contradicts the second.
In less moist forests (mean annual precipitation < 2000 mm), SIF was significantly positively correlated with precipitation and
significantly negatively correlated with VPD (Figs. S1 and S2). These relationships suggest that photosynthetic activity in
these regions are likely driven by both soil moisture availability and atmospheric water demand. On one hand, limited
precipitation reduces soil water content, constraining stomatal conductance and thereby lowering photosynthesis. On the other
hand, high VPD can prompt stomatal closure to mitigate water loss, which in turn decreases carbon uptake. Hence, the interplay
between precipitation and VPD jointly regulates photosynthetic fluxes, and both factors should be considered when interpreting
SIF variations in less moist forests.
Examining relationships across all ecoregions, we observed that as mean annual precipitation and the variability of monthly
total precipitation increased, the correlation between SIF and precipitation weakened, while the correlation between SIF and
VPD strengthened (became more negative) (Fig. 4). This indicates that in moist forests (mean annual precipitation > 2000 mm),
SIF was more strongly related to VPD than to precipitation.
Contrary to the second part of our hypothesis, we found that SIF and VPD were negatively correlated in moist forests rather
than positively correlated. This suggests that in these high-rainfall ecosystems, high atmospheric dryness (high VPD) inhibits
photosynthesis, leading to decreased SIF.
We also examined the relationships between vegetation indices (VIs) and environmental factors. Across all sites—except for
the Atlantic Equatorial Coastal Forest where relationships were not significant—the Enhanced Vegetation Index (EVI), Near-
Infrared Reflectance of vegetation (NIRv), and Normalized Difference Vegetation Index (NDVI) were negatively related to
VPD and positively associated with precipitation. The correlations between these VIs and temperature and photosynthetically
active radiation (PAR) were generally negative or insignificant.
Assessing whether SIF and the VIs were more strongly correlated with VPD or precipitation at each site, we found that
conclusions could depend on the correlation method used. For instance, in the Nigerian Lowland and Niger Delta Forest,
Spearman's correlation coefficient between SIF and VPD was –0.73, and between SIF and precipitation was 0.57. However,
using Pearson's correlation, the coefficients were –0.65 and 0.66, respectively. Thus, Spearman's correlation suggests that SIF
is more strongly correlated with VPD, while Pearson's indicates negligible differences between the correlations. To address
this discrepancy, we reported correlation matrices for both methods (Figs. S1 and S2).
To determine how relationships between SIF, EVI, and NDVI with VPD and precipitation changed with increasing mean
annual precipitation and variability in monthly precipitation, we compared correlation coefficients across all ecoregions.
Regardless of the correlation method used, the correlation between SIF and VPD strengthened (became more negative), while
the correlation between SIF and precipitation weakened with increasing mean annual precipitation and greater variability in
monthly precipitation (Fig. 4). This indicates that in forests with higher annual rainfall and more variable monthly precipitation,
SIF becomes increasingly related to VPD and less related to precipitation.
Conversely, NDVI showed a stronger correlation with precipitation and a weaker correlation with VPD in these wetter forests.
However, this relationship is likely influenced by NDVI's saturation in dense canopies, causing it to mirror the seasonality of
precipitation without capturing changes in photosynthetic activity. No significant correlation was found for EVI in these
forests.
When assessing the relationships between SIF, environmental factors, and VIs, we observed differing patterns in their
responses to precipitation and VPD across the ecoregions (Fig. 4). Specifically, while SIF showed a weakening correlation
with precipitation and a strengthening negative correlation with VPD as mean annual precipitation increased, EVI and NIRv
did not exhibit significant changes in their correlations with these environmental factors. NDVI displayed an opposite trend to
SIF, showing an increasing correlation with precipitation and a decreasing correlation with VPD in wetter forests.
These differing patterns suggest a potential decoupling between photosynthesis (as indicated by SIF) and canopy structure (as
indicated by VIs like EVI and NIRv) in African tropical forests. In moist forests with high mean annual precipitation and
variability, the canopy structure remains relatively constant throughout the year due to the evergreen nature of the forests. This
results in stable EVI and NIRv values that are less sensitive to short-term environmental fluctuations. In contrast,
photosynthetic activity, as indicated by SIF, can vary significantly in response to atmospheric conditions such as VPD.
One possible explanation for this decoupling is that SIF is more directly linked to the physiological status of leaves, capturing
changes in photosynthetic efficiency and electron transport rates not necessarily reflected in canopy structural metrics. High
VPD in moist forests can lead to stomatal closure to prevent excessive water loss, reducing $CO_2$ uptake and photosynthesis
without significantly altering canopy structure or leaf area index (LAI). As a result, SIF decreases while EVI and NIRv remain
relatively unchanged.
In less moist forests, where soil moisture deficits are more common, both photosynthesis and canopy structure can be affected
by changes in precipitation. Limited water availability can lead to reduced leaf area through leaf shedding or inhibited growth,
which is reflected in decreases in both SIF and VIs. This explains the stronger coupling between SIF and VIs in response to
precipitation in these regions.


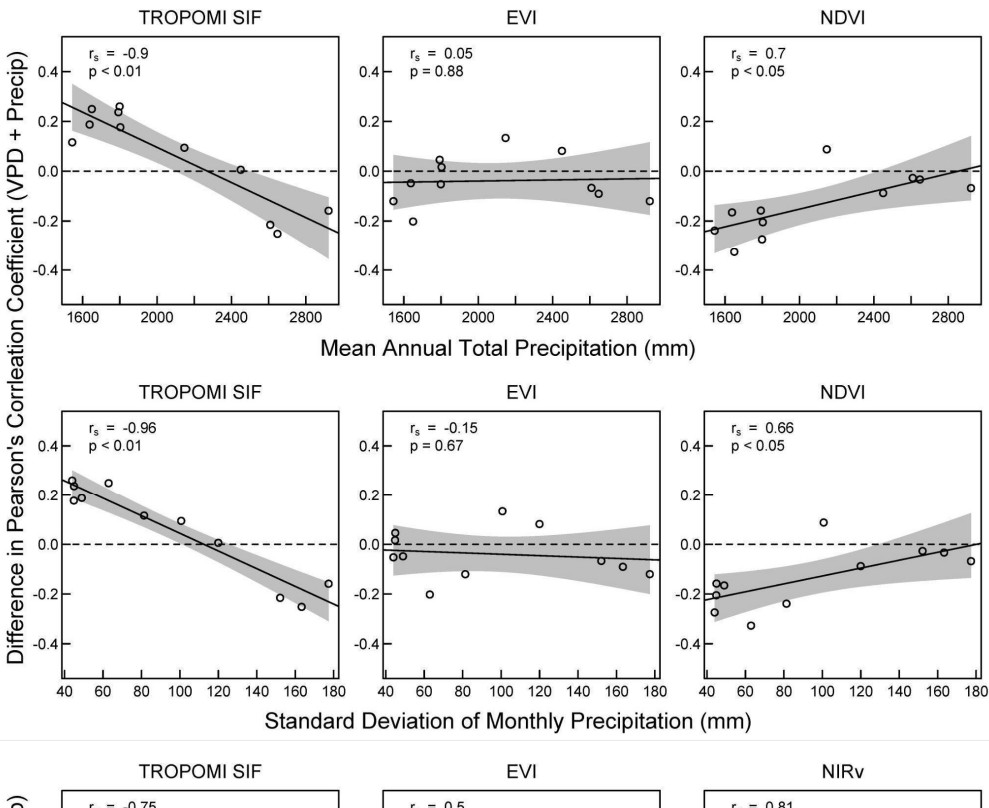


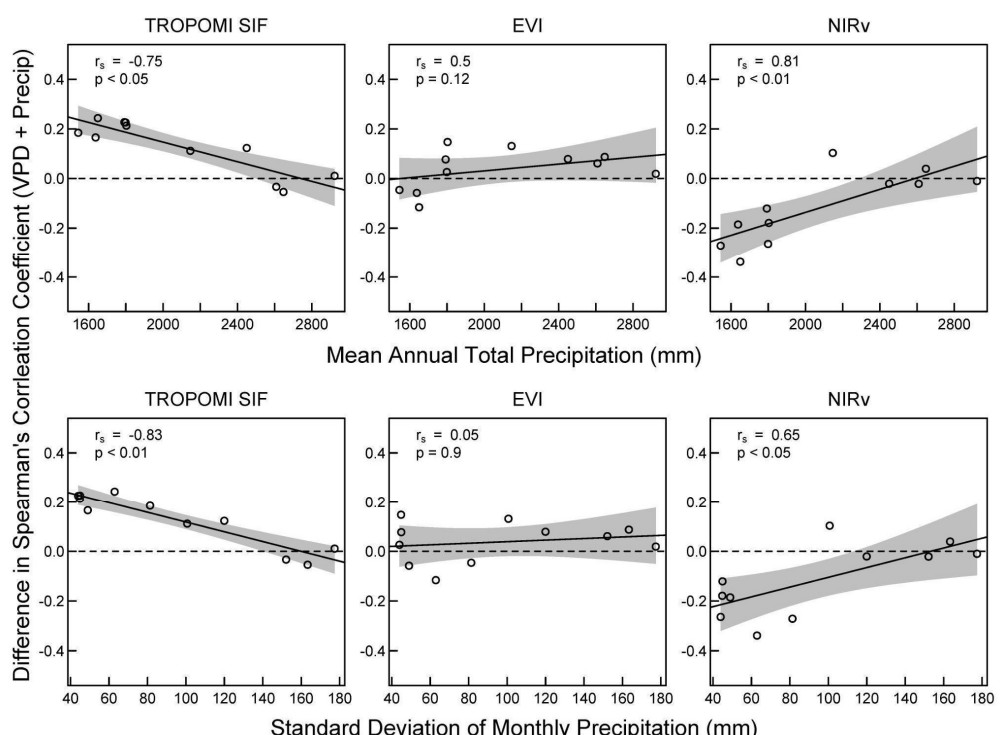


**Figure 4. Regressions of the differences in the correlation coefficients for SIF, EVI, and NDVI vs VPD and precipitation using**
**Pearson's and Spearman's correlation tests for each ecoregion. Differences are *r* for VPD plus    *r* for precipitation (note that r for**
**VPD is always negative and r for precipitation always positive). Top two rows are differences in Pearson's correlation coefficient,**
**and bottom two rows are differences in Spearman's correlation coefficient.**
**3.4    Synchrony in minimum and maximum SIF, EVI, and environmental factors**
The ecoregions of Central Africa straddle the equator, so we evaluated whether the double-peak seasonality in SIF, EVI, and
environmental factors were occurring locally, or whether the double peaks that we observed at the ecoregion scale were due
to the peaks alternating in time between the northern and southern regions. For SIF and EVI, we found that the double-peak
seasonality was largely a continental-scale phenomenon (Fig. 5, top two rows). Except for some forests at the northern and
southern most fringes, Central African tropical forests exhibited a double-peak seasonality in SIF and EVI.

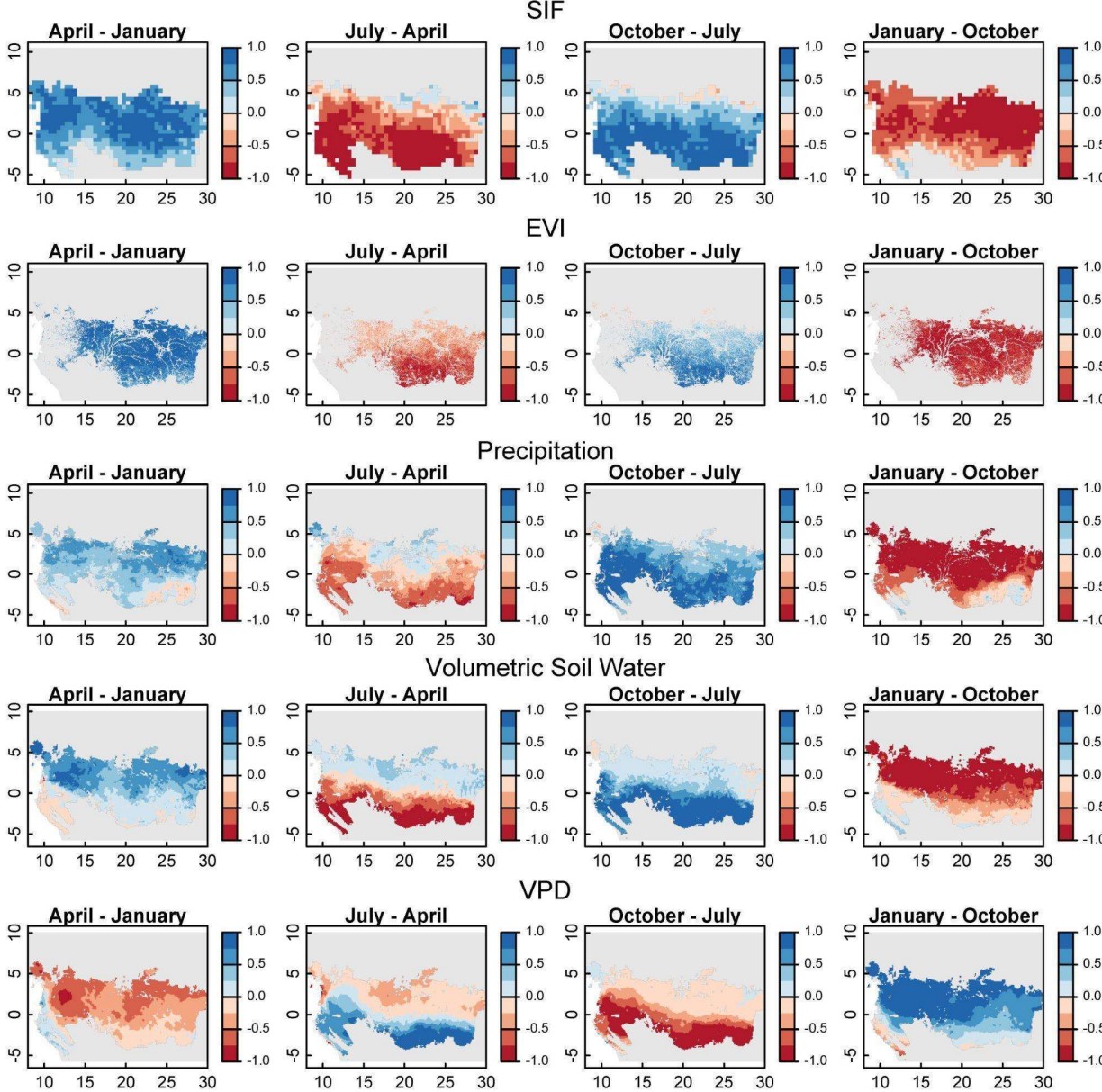

**Fig. 5. Differences in normalized SIF, EVI, precipitation, volumetric soil water, and VPD. Shown are the changes from January to April (first column), April to July (second column), July to October (third column), and October to January (fourth column). The periods chosen correspond to the double-peak seasonality of precipitation for these tropical forests: January to April (increasing precipitation), April to July (decreasing precipitation), July to October (increasing precipitation), and October to January (decreasing precipitation).**

For precipitation, soil moisture, and VPD, the double-peak seasonality was mostly constrained to the southern portions of the Central African forest (Fig. 5, bottom three rows), with the northern regions more commonly experiencing increasing precipitation from January to April, April to July, and July to October with a single decrease occurring between October and

January. These different precipitation regimes for the northern and southern regions of the Central Africa forests create a clear
north-south divide in the timing of minimum SIF, EVI, precipitation, and soil moisture and maximum VPD (Fig. 6, first
column). In the northern region, these minimums and maximums in VPD occur in the beginning of the calendar year following
the southern solstice, and in the southern region they coincide with the mid-year northern solstice.
Interestingly, we did not find a similar bimodal north-south timing in maximum SIF, EVI, precipitation, soil moisture, or
minimum VPD. Maximum SIF and EVI had bimodal distributions, but these distributions did not have clear geographical
patterns. Maximum precipitation, soil moisture, and minimum VPD occurred nearly exclusively in the last quarter of the year.

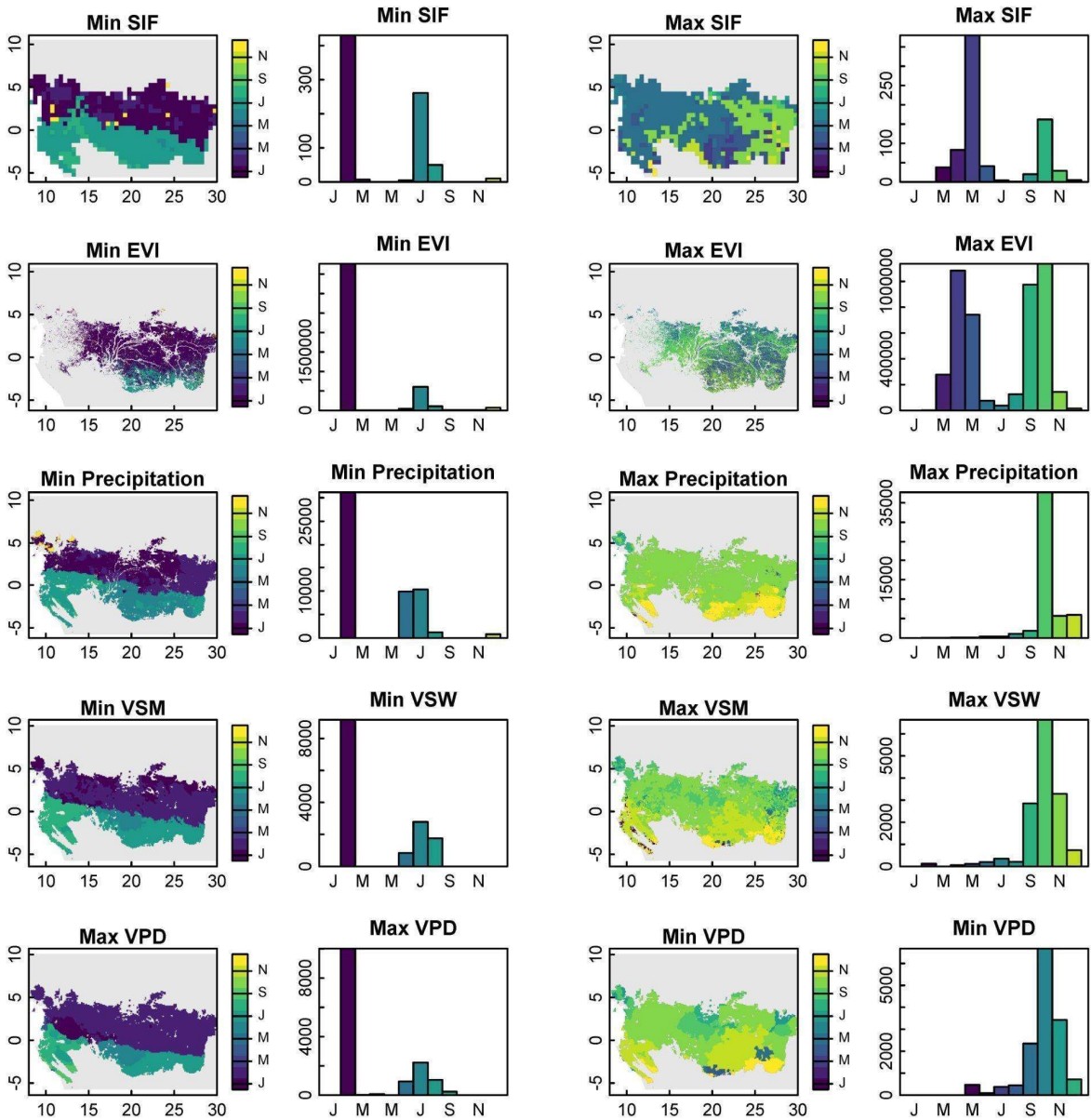


**Fig. 6. Month in which minimum and maximum SIF, EVI, precipitation, volumetric soil water, and VPD occur in tropical African**
**forests. For columns 2 and 4, the x-axis is month and the y-axis is the number of grid cells.**

## 4 Discussion

### 4.1 Contrasting SIF dynamics and precipitation regimes in West African and Amazonian moist tropical forests

Here, we situate our findings within the broader context of what is already known about other moist tropical forests—particularly those in the Amazon—to discern universal patterns and region-specific dynamics. This comparative perspective enhances our comprehension of how different moist tropical forests respond to variations in precipitation, vapor pressure deficit (VPD), and photosynthetically active radiation (PAR), thereby contributing to more accurate models of global tropical forest behavior under changing environmental conditions.

There were stark similarities in the timing of SIF, VIs, and precipitation in the moist Amazon and West African moist African forests, despite their apparent opposing responses to VPD. For both forest types, SIF and the VIs increased as precipitation increased after a distinct two- or three-month dry period, and SIF and the VIs peaked prior to peak precipitation and as $PAR_{TOC}$ declined. These comparisons indicate that there are pan-tropical similarities in leaf physiology and demography in both the moist Amazon and West African moist tropical forests, with new leaf flush and increased productivity as the dry season is alleviated by increased rainfall.

However, we observed three main differences in the precipitation regimes of these West African moist forests and the Amazon forests. First, annual peak monthly precipitation for West African moist tropical forests was frequently as high as 400 mm - 600 mm, much higher than what has been observed in moist Amazon forests. These periods of intense rainfall in Africa cause reductions in $PAR_{TOC}$ that are much larger than those seen in the Amazon, and may suppress $APAR_{chl}$, photosynthesis, and SIF as the ecosystem becomes light-limited. The alleviation of this light limitation in the weeks after peak precipitation could explain the second peak in SIF despite any noticeable changes in the VIs.

Second, the distribution of rainfall in a year was sometimes bimodal for the West African moist tropical forests, particularly for the Nigerian Lowland and Niger Delta Forest and the Western Guinean Lowland Forest, which likely contributed to the variability seen in the monthly SIF data for these ecoregions. Conversely, the precipitation regimes in the Amazon have been reported to be much less variable and are normally distributed (Liang et al., 2020).

Third, minimum precipitation in West African moist tropical forests occurs during the southern solstice around December, when the solar zenith angle and diffuse radiation is highest for that region, and maximum precipitation occurs between the northern solstice and southward equinox when PAR at the top of the atmosphere ($PAR_{TOA}$) is highest. In contrast, minimum and maximum precipitation for moist Amazon forests occurs during the southward and northward equinox, respectively.

Although we confirmed our first hypothesis that SIF in more moist ecoregions is less related to precipitation, we were surprised to find that SIF had a more negative relationship with PAR and VPD in West African moist tropical forests than those in Central Africa. We expected these forests to be radiation-limited rather than water-limited due to their high mean annual total rainfall. Instead, we found substantial decreases in SIF that were synchronous with increased PAR and VPD immediately following the wet season. These observations were contrary to what has been observed in the moist forests of the Amazon, where positive relationships exist between SIF, photosynthesis, and VPD due to newly flushed leaves during the dry season,

which have higher photosynthetic capacity and compensate for stomatal closure due to increased VPD (Green et al., 2020; Wu
et al., 2016). For West African moist tropical forest, our findings indicate that leaf abscission and leaf flush are synchronous
with increasing and decreasing VPD.

**4.2 Central African tropical forest**

Correlative analyses of SIF, EVI, and environmental factors for Central African tropical forests at the continental or
ecoregional scale indicate that SIF and EVI are synchronous with precipitation (Fig. 1). Such synchrony and high correlations
allude that SIF and EVI, and thus the productivity of Central African tropical forests, are driven by precipitation. However,
the drivers of the seasonality of SIF and EVI are much more complex.
For instance, spaceborne-observed physiology (SIF) and phenology (EVI) of the entire Central African tropical forest region
acts in concert, with a bimodal seasonality in SIF and EVI that occurs in lockstep across the entire continent (Fig. 5, first two
rows). This continental phenomenon occurs despite north-to-south differences in precipitation, soil moisture, and VPD regimes
(Fig. 5, last three rows). Studies have shown that Central African tropical forests are extraordinarily resistant to precipitation
anomalies (Asefi-Najafabady and Saatchi, 2013), long-term declines in annual total precipitation (Jiang et al., 2019; Malhi and
Wright, 2004; Sun et al., 2022), and anomalies in the El Niño–Southern Oscillation and Madden–Julian Oscillation (Bennett
et al., 2021; Raghavendra et al., 2020). Thus, the continental-scale bimodal seasonality of SIF and EVI are likely more related
to solar insolation and angle than climate (Calle et al., 2010; van Schaik et al., 1993).
Nevertheless, the timing of minimum SIF and EVI are synchronous with minimum precipitation (Fig. 6, first two columns).
Given the numerous studies that have highlighted the insensitivity of Central African tropical forests to precipitation anomalies,
it is likely that the minimums in precipitation and solstice-related maximums in solar zenith angle serve as phenological queues
for leaf abscission rather than directly inducing water-stress related declines in productivity. This explanation is supported by
field observations, which found a north-south bimodal timing in peak leaf flush Central African tropical forests (van Schaik et
al., 1993). Similarly, the timing of maximum SIF and EVI are likely less related to environmental factors (Fig. 6, last two
columns) and could be more related to localized characteristics, such as herbivory, disturbance, topography, species
composition, forest structure and age, soil characteristics, or other potential biotic and abiotic factors.

**4.3 Tropical rainforest with relatively low annual rainfall**

Our study encompassed several tropical forest ecoregions with mean annual precipitation at or below 1700 mm, such as the
Eastern Guinean, Western Congolian Swamp Forest, and Northwestern Congolian Lowland Forest (Figure 1). This
precipitation level is at the lower end for sustaining tropical forests. The fact that these forests remain green year-round despite
lower rainfall suggests that they possess unique adaptations to cope with periodic water limitations.
The lower annual precipitation in these ecoregions is reflected in their physiological and phenological responses. We observed
that SIF and vegetation indices like EVI and NDVI in these regions showed a strong synchrony with precipitation and VPD
(Figs. S2 and S3). This suggests that in drier evergreen forests, water availability becomes a more significant driver of
photosynthesis seasonality compared to wetter regions. The strong negative correlations between SIF and VPD, and positive
correlations with precipitation, indicate that photosynthetic activity in these ecoregions is more sensitive to atmospheric
dryness and water stress. This sensitivity could be due to several factors.
First, trees in drier evergreen forests may have evolved mechanisms to optimize water use efficiency, such as closing stomata
during periods of high VPD to reduce transpiration losses, which in turn affects photosynthesis (Eamus and Prior, 2001).
Second, these forests might rely on deep root systems to access groundwater during dry periods, but prolonged low
precipitation can still lead to water deficits affecting canopy function (Schenk and Jackson, 2002). Finally, the tree species in
these regions may have traits that are adapted to lower water availability, such as being semi-deciduous, influencing the overall
ecosystem response to environmental drivers (Pennington et al, 2009).
By including ecoregions with lower annual precipitation, our study captures a broader spectrum of tropical forest responses to
environmental factors. It highlights that even within evergreen tropical forests, there is significant variability in how
ecosystems respond to changes in precipitation and atmospheric conditions. This variability underscores the importance of
considering local precipitation regimes when interpreting remote sensing signals like SIF and vegetation indices.
**4.4 Robustness of the SIF signal**
OCO-2 and OCO-3 agreed well with TROPOMI across an aggregate of the ecoregions, and the OCOs agree well with each
other (Figs. 7 and S3). At the ecoregion level, the OCOs can better capture the seasonality of TROPOMI SIF in ecoregions
with large intact forests due to a larger volume of OCO data at larger spatial scales, albeit with a slight high bias (Fig. S4 and
S5). The higher bias of the OCO-2 and OCO-3 retrievals relative to TROPOMI could be related to the nadir viewing angle of
the OCOs and differences in the retrieval windows, instrument characteristics, bias correction, and/or footprint size.

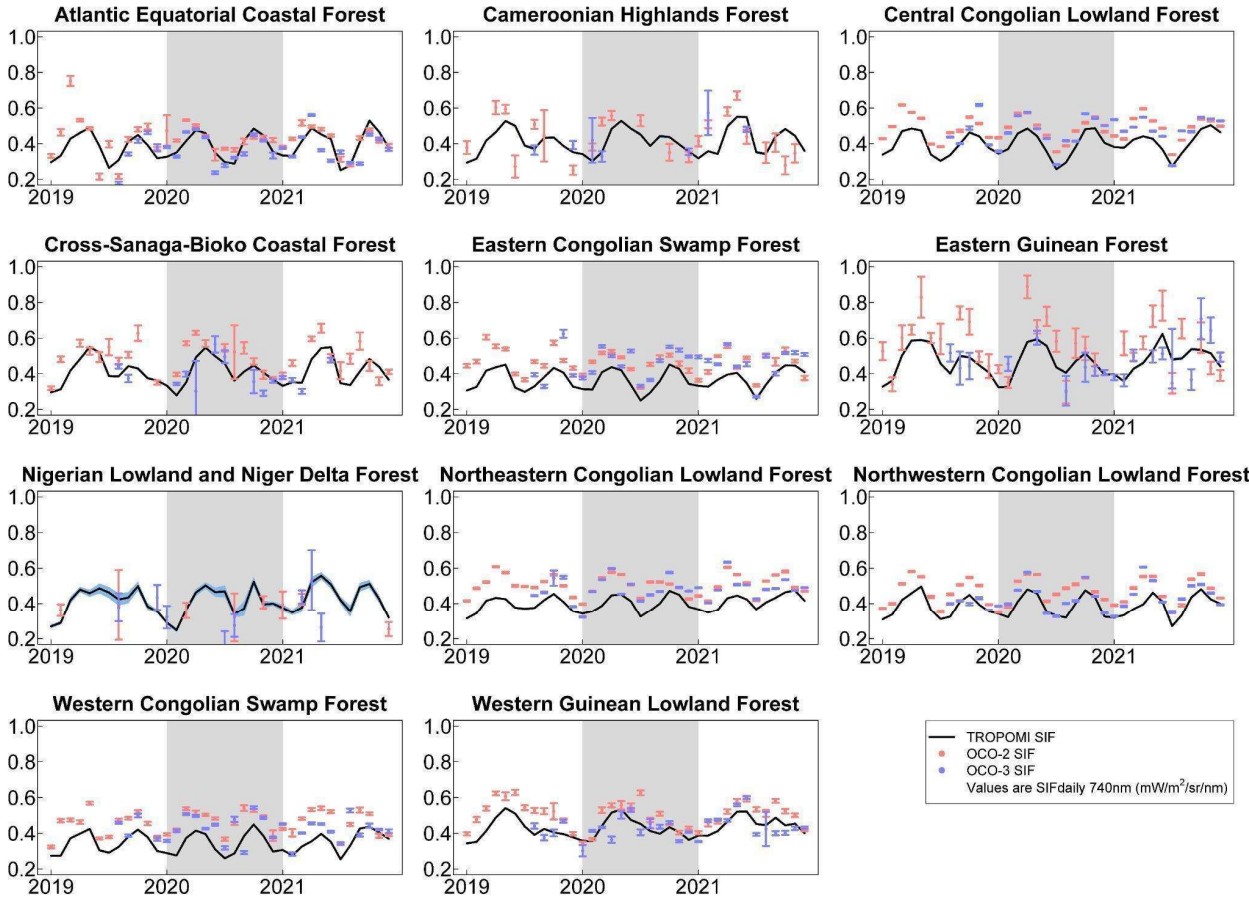


Fig 7. Monthly mean SIF from TROPOMI, OCO-2, and OCO-3 for 11 African tropical ecoregions for 2019 - 2021. OCO-2 and OCO-3 error bars and blue shaded region under the TROPOMI line are +/- the standard error of the mean. The shaded region delineates the year 2020. Values are SIFdaily 740nm (mW/m2/sr/nm).

**4.4 Uncertainties and Limitations**

While our study advances the understanding of photosynthetic seasonality in African tropical forests using spaceborne SIF data, we acknowledge that there are several uncertainties and limitations of our study.

First, satellite-based SIF measurements, although powerful for large-scale observations, come with inherent uncertainties due to sensor characteristics and retrieval algorithms. Differences between OCO-2, OCO-3, and TROPOMI—such as spatial resolution, overpass time, viewing geometry, and retrieval methods—can introduce biases and inconsistencies in the SIF data (Doughty et al., 2021a). The higher bias observed in OCO-2 and OCO-3 relative to TROPOMI may be attributed to these factors, potentially affecting the comparability of our results across different sensors.

Second, atmospheric conditions prevalent in tropical regions, particularly cloud cover and aerosols, can impact the accuracy of SIF retrievals despite efforts to mitigate these effects (Guanter et al., 2015). Although we avoided applying a cloud fraction

threshold to prevent clear sky bias, residual atmospheric disturbances may still influence the SIF signals, leading to possible
overestimation or underestimation of photosynthetic activity.
Third, the lack of ground-based validation data poses a significant limitation. The scarcity of eddy covariance towers and other
in situ measurements in structurally intact African tropical forests restricts our ability to validate satellite-derived SIF data and
to calibrate the relationships between SIF, GPP, and environmental variables (Malhi, 2012; Williams et al., 2007). Without
ground truthing, it remains challenging to disentangle the contributions of leaf physiology, phenology, and canopy structure
to the observed SIF signals, and to attribute observed variations in SIF to specific environmental drivers with high confidence.
Fourth, our assumption that SIF is a reliable proxy for photosynthetic activity may not fully capture the complexity of plant
physiological responses. SIF is influenced by multiple factors, including canopy structure, leaf area index, chlorophyll content,
and fluorescence yield, which can vary independently of GPP (Porcar-Castell et al., 2014; Magney et al., 2020). The
relationship between SIF and GPP can be affected by environmental stressors, such as VPD and temperature, which may alter
fluorescence efficiency without a proportional change in photosynthesis (Gonçalves et al., 2020). Without concurrent
measurements of leaf-level physiological parameters, attributing changes in SIF to specific processes remains challenging.
Additionally, the climate and environmental data used in our analyses introduce their own uncertainties. Reanalysis products
like ERA5 may not capture localized microclimatic variations and can be less accurate in regions with sparse observational
data (Beck et al., 2017). Satellite-derived vegetation indices may suffer from saturation effects in dense canopies and can be
influenced by sensor noise and atmospheric conditions (Huete et al., 1997b). These uncertainties may affect our assessments
of the relationships between SIF, vegetation indices, and environmental factors.
Furthermore, our spatial analysis at the ecoregion level may mask heterogeneity within ecoregions due to variations in species
composition, soil properties, topography, and anthropogenic disturbances. The coarse spatial resolution of some datasets may
lead to mixed pixels, especially near forest edges or in fragmented landscapes, potentially confounding the interpretation of
SIF signals.
Addressing these limitations requires the integration of additional ground-based observations, improved satellite retrieval
algorithms, higher-resolution datasets, and extended time series. Establishing a network of eddy covariance towers and
phenological monitoring sites across African tropical forests would greatly enhance the validation and interpretation of
satellite-derived SIF data. Future studies should also explore advanced modeling approaches that account for the complexity
of plant physiological processes and incorporate data from upcoming satellite missions with enhanced capabilities.
**4.5    Future work**
In deciduous ecosystems, SIF, photosynthesis, and vegetation indices are typically well correlated because both are driven by
strong seasonalities in leaf area, canopy chlorophyll, and phenology (Doughty et al., 2021b). However, in evergreen
ecosystems, including boreal needleleaf and tropical broadleaf, SIF and GPP can exhibit seasonal dynamics that do not well
match vegetation indices because SIF and GPP are sensitive to changes in leaf demography, leaf physiology, and $APAR_{chl}$
(Doughty et al., 2021b, 2019; Gonçalves et al., 2020; Pierrat et al., 2022). For instance, it was found that spaceborne SIF had
very low to no significant correlation with EVI and NDVI across tropical evergreen broadleaf forest in South America, Africa,
and Southeast Asia (Doughty et al., 2021b), but that the seasonality of SIF in the Amazon and Africa well matched eddy
covariance tower GPP (Doughty et al., 2019; Mengistu et al., 2021).
Anecdotally, we would expect the seasonality of SIF to also well match GPP in African tropical evergreen broadleaf forests,
but such a comparison is not possible due to the absence of eddy towers in in-tact African tropical forests that are large enough
to not cause mixed-pixel effects with high temporal resolution satellite data. Ideally, a future eddy tower network in Africa
would be established in tropical forests across a precipitation gradient and include West African moist tropical forests and
Central African tropical forests. Litter-fall traps and phenology cameras at the tower sites would enable us to determine to
what degree observed changes in GPP are attributable to climate and leaf demography and physiology and could be directly
compared to spaceborne SIF and surface reflectance to validate our satellite observations.
Another topic that can be addressed is whether there is a significant long-term trend in SIF in the Central African tropical
forests, as it has been debated whether there is a browning trend in these forests (Sun et al., 2022; Zhou et al., 2014). As the
global spaceborne SIF record continues to lengthen over time, analyses of the SIF data records will allow us to not only address
whether there are long-term changes in greenness or leaf area but provide a clue as to whether there is a long-term trend in
photosynthesis. Similarly, the SIF data record has not yet been levered to address whether anomalies in SIF occur during
periods of La Nina or El Nino or during precipitation anomalies.
**Data availability**
All data used in our paper is publicly available. OCO-2 SIF Lite files can be accessed at
https://doi.org/10.5067/XO2LBBNPO010 (OCO-2 Science Team et al., 2020), and OCO-3 data can be accessed at
https://doi.org/10.5067/NOD1DPPBCXSO (OCO-3 Science Team et al., 2020). TROPOMI SIF can be accessed at
https://ftp.sron.nl/open-access-data-2/TROPOMI/tropomi/sif/v2.1/l2b/. CHIRPS Precipitation can be accessed at
https://data.chc.ucsb.edu/products/CHIRPS-2.0/. ERA-5 Reanalysis can be found at https://cds.climate.copernicus.eu/.
MODIS MCD43A4 data can be accessed at https://lpdaac.usgs.gov/products/mcd43a4v061/. Copernicus forest cover can be
accessed at https://land.copernicus.eu/global/products/lc. Terrestrial ecoregions can be accessed at
https://www.worldwildlife.org/publications/terrestrial-ecoregions-of-the-world.
**Author contributions**
RD and MW conceived this paper. RD and DW performed the analyses. RD prepared the paper with contributions from all
co-authors.

## Competing interests

The contact author has declared that neither they nor their co-authors have any competing interests.

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
