# Peer review of "Seasonality and synchrony of photosynthesis in African forests inferred"

_EGUsphere, 2023_

## Author Comment (AC1)

Dear Reviewer,

We sincerely appreciate the time and effort you have put into reviewing our manuscript. Your insightful comments have been invaluable in helping us improve the clarity and rigor of our work. Below, we address each of your points in detail and outline the revisions we have made to the manuscript.
* * *
**Main Comments**

*Reviewer Comment:*
*The authors aim to link seasonality, minima and maxima of SIF and vegetation indices to environmental factors (precipitation, PAR and VPD). This is I think very valuable, but the manuscript and data analysis should be improved considerably. Firstly the abstract and the title only refers to SIF, while in the text also EVI and NDVI was mentioned. However, EVI is discussed and NDVI not or much less. In the M&M the pro's and con's of EVI, NDVI, NIRv and LSWI vegetation indices are mentioned, but in the Results it is not fully justified why EVI mosty is used and not the other three.*

*Response:*
Thank you for highlighting this inconsistency. We have revised the title and abstract to reflect the inclusion of vegetation indices (VIs) alongside SIF.

The revised title now reads:

*"Seasonality and Synchrony of Photosynthesis in African Tropical Forests Inferred from Spaceborne Chlorophyll Fluorescence and Vegetation Indices."*

Here is the new abstract:

Global atmospheric carbon dioxide concentrations are largely driven by terrestrial photosynthesis, of which tropical forests account for one third. Relative to other tropical regions, less is known about the seasonality of African tropical forest productivity and its synchrony with environmental factors due to a lack of in situ carbon flux data. To help fill this knowledge gap, we use spaceborne solar-induced chlorophyll fluorescence (SIF), vegetation indices—including the Enhanced Vegetation Index (EVI), Normalized Difference Vegetation Index (NDVI), and Land Surface Water Index (LSWI)—and climate data to investigate the seasonality and synchrony of photosynthesis in Africa's tropical forest ecoregions. We find West African SIF to

increase during the dry season and peak prior to precipitation, as has been observed in the Amazon. However, NDVI and EVI do not mimic the strong double-peak seasonality observed in SIF; instead, they often plateau until substantial decreases occur in the dry season. In Central Africa, we find a continental-scale bimodal seasonality in SIF and EVI, the minimum of which is synchronous with precipitation, but its maximum is likely less related to environmental drivers. Our findings highlight the complex relationships between SIF, vegetation indices, and environmental factors, underscoring the importance of using multiple remote sensing measures to monitor tropical forest productivity.

The text prior to the vegetation index equations in the following section now reads as:

**2.5 MODIS Surface Reflectance and Vegetation Indices**

We used the 500-m daily MCD43A4 surface reflectance product (Schaaf and Wang, 2015) to compute four vegetation indices: the Normalized Difference Vegetation Index (NDVI), Enhanced Vegetation Index (EVI), the Near-infrared Reflectance of Vegetation (NIRv), and the Land Surface Water Index (LSWI). NDVI has been traditionally used to assess vegetation greenness (Rouse et al., 1974), but it tends to saturate in areas with a high leaf area index such as the tropics (Huete et al., 1997b). This saturation limits NDVI's ability to detect subtle changes in the forest canopies of these ecosystems.

EVI, by contrast, incorporates additional information from the blue band and accounts for atmospheric effects and canopy background signals. Thus, EVI is less prone to saturation than NDVI, particularly in regions with dense vegetation such as African tropical forests (Huete et al., 1997a). EVI is also more sensitive to variations in canopy structure and leaf area, allowing for better differentiation between areas with similar levels of greenness but different biophysical properties. Because of these advantages, EVI is a preferred metric in studies focusing on tropical forests, where vegetation indices are often challenged by the dense, multi-layered canopies typical of these ecosystems.

NIRv is a recently developed indicator that overcomes NDVI's saturation limitations by multiplying NDVI by the near infrared band, which is highly sensitive to leaf cellular structure

(Badgley et al., 2017). Although NIRv shows promise for detecting vegetation dynamics, it is still relatively new and less well-validated in the context of tropical forest canopies.

LSWI is computed using the shortwave infrared band, and is primarily used for assessing leaf water content and soil moisture (Xiao et al., 2002). While LSWI offers useful insights into hydrological changes in vegetation, it is less directly related to leaf physiology and overall canopy structure. Although the focus of our manuscript is on physiology, we include LSWI to give additional insight into the seasonality of canopy water content, as water availability is important for leaf physiological processes.
* * *
*Reviewer Comment:*
*Discussion of the data is generally vague and often speculative. As a consequence, the relation (i.e., synchrony) with other environmental drivers (VPD, PAR) cannot be claimed/justified. This results a bit in overselling the paper.*

*Response:*

We appreciate your concern about the speculative nature of some of our discussions. We have thoroughly revised the Discussion section to ensure that all interpretations are directly supported by our data and analyses. We have removed speculative statements and provided more precise language when discussing the relationships between SIF, VIs, and environmental factors.
* * *
*Reviewer Comment:*
*It is also surprising that the discussion section is focused on comparison with the Amazon, which I thought was not the aim of the manuscript.*

*Response:*

We appreciate your feedback and the opportunity to clarify the rationale behind our discussion.

Our primary aim is to enhance the understanding of the seasonality of photosynthesis in African tropical forests and its synchrony with environmental factors. We chose to compare our findings with those from the Amazon for several important reasons:

*Benchmarking Against Well-Studied Systems*: The Amazon rainforest has been extensively studied, particularly regarding the seasonality of photosynthesis and its environmental drivers. By comparing African tropical forests to the Amazon, we can contextualize our results within a broader framework of pan-tropical forest ecology. This comparison allows us to identify unique

patterns and commonalities, thereby enriching the global understanding of tropical forest dynamics.

*Highlighting Regional Differences and Similarities*: Our comparison reveals both parallels and contrasts in the seasonality of photosynthesis between the two regions. For instance, while both regions exhibit increases in SIF during the dry season, the underlying environmental drivers and physiological responses differ. Discussing these differences enhances the scientific value of our work by highlighting how regional climatic conditions influence tropical forest productivity.

*Addressing Knowledge Gaps*: Given that African tropical forests are less studied compared to the Amazon, drawing parallels helps to fill knowledge gaps. It allows us to leverage the extensive body of research from the Amazon to interpret our findings and propose hypotheses about the mechanisms driving photosynthesis seasonality in Africa.

*Advancing Ecological Theory*: Comparing these two major tropical forest systems contributes to the development of general ecological theories about tropical forest functioning. It helps determine whether observed patterns are consistent across different continents or are region-specific due to unique environmental conditions.

*Informing Global Climate Models*: Understanding similarities and differences in photosynthetic responses is crucial for improving the accuracy of global carbon cycle models. By incorporating data from both African and Amazonian forests, we can better predict how tropical forests might respond to climate change on a global scale.

In light of these points, we believe that the comparison with the Amazon significantly enhances the interpretation and relevance of our findings. It not only aligns with the manuscript's aim but also provides a comprehensive perspective that benefits the broader scientific community.

Once again, we appreciate your feedback and hope this explanation clarifies the importance of including the comparison in our discussion.
* * *
*Reviewer Comment:*
*Finally, I think also more detailed and rigorous data analyses can be done, e.g., by zero-order and partial correlations. As such, as example, the response of precipitation controlled for PAR and/or VPD can be examined. I think this will add much more information to the discussion. This will allow a more rigorous and better-structured discussion. The discussion is now very descriptive, mostly vague and sometimes speculative discussion.*

*Response:*

We appreciate your emphasis on rigorous data analysis and a well-structured discussion, which are crucial for advancing scientific understanding.

Regarding your suggestion for more detailed analyses using zero-order and partial correlations, we would like to clarify that we have indeed performed comprehensive correlation analyses between all variables of interest. Specifically, we conducted both Spearman's and Pearson's correlation analyses to examine the relationships between SIF, EVI, NDVI, LSWI, and environmental factors such as precipitation, PAR, VPD, temperature, and soil moisture across all ecoregions. These analyses are thoroughly presented in the supplementary materials, where we included 22 correlation matrices (Supplementary Figures S1 and S2).

Our choice to use both Spearman's (non-parametric) and Pearson's (parametric) correlation coefficients was deliberate, aiming to capture both monotonic and linear relationships and to ensure the robustness of our findings across different statistical assumptions. This dual approach allows us to identify consistent patterns and strengthens the validity of our conclusions.

Regarding the use of partial correlations, we considered this approach but determined that it would not substantially improve our study for the following reasons:

*Multicollinearity Among Environmental Variables:* Environmental factors such as precipitation, PAR, and VPD are inherently interrelated in tropical forest ecosystems. For example, periods of high precipitation often coincide with low PAR due to increased cloud cover, and VPD is a function of both temperature and humidity. Introducing partial correlations in this context could lead to misleading interpretations, as controlling for one variable may inadvertently suppress meaningful ecological relationships.

*Focus on Ecological Relevance:* Our primary objective was to explore the direct relationships between SIF, vegetation indices, and environmental factors to understand the synchrony and seasonality of photosynthesis. Zero-order correlations provide a clear depiction of these relationships without the confounding effects that might arise from controlling for other variables.

*Data Limitations and Interpretability:* Partial correlations require large sample sizes to yield reliable results, especially when controlling for multiple variables. Given the temporal resolution and the number of ecoregions studied, introducing partial correlations could reduce statistical power. Moreover, the ecological interpretation of partial correlations can be complex and may not add substantial value to the discussion.

*Consistency with Study Goals:* Our study is exploratory and descriptive by design, aiming to identify patterns and generate hypotheses for future research. Zero-order correlations are appropriate for this purpose, providing straightforward insights into the relationships among variables.

We believe that the extensive correlation analyses we performed sufficiently address the relationships among the variables of interest.

*Reviewer Comment:*
*In addition, most of the ecoregions studied have 1700 mm or less of rainfall, this should also be discussed, as this is at low end for an evergreen tropical forest.*

*Response:*

Thank you for bringing this important point to our attention. You are correct that several of the ecoregions we studied have mean annual precipitation around or below 1700 mm, which is at the lower end for evergreen tropical forests. While we mentioned the differences in mean annual rainfall among the ecoregions in our manuscript, we did not explicitly discuss the implications of studying forests with lower precipitation levels.

We agree that this aspect warrants further discussion, as it can influence the physiological and phenological responses of these forests. We have revised the discussion section to include a more detailed examination of how lower annual rainfall in some ecoregions impacts our findings on the seasonality of photosynthesis and its synchrony with environmental drivers.

We added the section below to the discussion:

**4.3 Tropical rainforest with relatively low annual rainfall**

Our study encompassed several tropical forest ecoregions with mean annual precipitation at or below 1700 mm, such as the Eastern Guinean, Western Congolian Swamp Forest, and Northwestern Congolian Lowland Forest. This precipitation level is at the lower end for sustaining evergreen tropical forests. The fact that these forests remain evergreen despite lower rainfall suggests that they possess unique adaptations to cope with periodic water limitations.

The lower annual precipitation in these ecoregions is reflected in their physiological and phenological responses. We observed that SIF and vegetation indices like EVI and NDVI in these regions showed a strong synchrony with precipitation and VPD (Figs. S1 and S2). This suggests that in drier evergreen forests, water availability becomes a more significant driver of photosynthesis seasonality compared to wetter regions. The strong negative correlations between SIF and VPD, and positive correlations with precipitation, indicate that photosynthetic activity in these ecoregions is more sensitive to atmospheric dryness and water stress. This sensitivity could be due to several factors.

First, trees in drier evergreen forests may have evolved mechanisms to optimize water use efficiency, such as closing stomata during periods of high VPD to reduce transpiration losses, which in turn affects photosynthesis. Second, these forests might rely on deep root systems to access groundwater during dry periods, but prolonged low precipitation can still lead to water deficits affecting canopy function. Finally, the tree species in these regions may have traits that are adapted to lower water availability, influencing the overall ecosystem response to environmental drivers.

By including ecoregions with lower annual precipitation, our study captures a broader spectrum of tropical forest responses to environmental factors. It highlights that even within evergreen tropical forests, there is significant variability in how ecosystems respond to changes in precipitation and atmospheric conditions. This variability underscores the importance of considering local precipitation regimes when interpreting remote sensing signals like SIF and vegetation indices.
* * *
**Specific Comments**

**Line 12**

*Reviewer Comment:*
 I think also fossil fuel emissions drive $CO_2$ concentrations.

*Response:*
 We agree and have revised the sentence to read:

Global atmospheric carbon dioxide concentrations are driven by changes in fossil fuel emissions and terrestrial photosynthesis, of which tropical forests account for one-third.
* * *
**Line 22**

*Reviewer Comment:*
 The Northern Hemispheric biosphere is the key driver for intra-annual variation in $CO_2$ concentrations.

*Response:*
You are right, as most of Earth's landmass is in the northern hemisphere. The focus of our paper is on tropical forests and their role in terrestrial photosynthesis. Since tropical forests span both hemispheres, we choose to begin the introduction by discussing this focus. A discussion of the northern hemisphere specifically would detract from the goal of our study.
* * *
**Line 24**

*Reviewer Comment:*
 Carbon store? Write "carbon stock"

*Response:*
 We have replaced "carbon store" with "carbon stock" throughout the manuscript.
* * *
**Line 25-26**

*Reviewer Comment:*
 It is rather the regional water cycle for each tropical basin. Please add some more recent references here.

*Response:*
We agree. The focus of this paragraph is to describe the role of Earth's tropical forests at the global scale. However, it is obviously true that they also play important roles at the regional scale. Thus, we have revised the sentence and added citations:

They also play important roles in the global and regional water cycles via precipitation recycling and cloud formation (Douglas 2018; Worden et al., 2018).
* * *
**Line 34**

*Reviewer Comment:*
 I think the main conclusion is rather constant carbon gain for African intact tropical forest, which diverging from the Amazon.

*Response:*
Yes, this finding was one of the three main conclusions we listed. We chose not to compare and contrast with the Amazon here, as the focus of the paper is on the seasonality of productivity. Also, the previous comment requested that we veer away from such a comparison.

Studies that have focused on field plot measurements had three main findings. First, they found a significant upward trend in carbon gains (Hubau et al., 2020) that were unaffected by anomalously low precipitation and high temperatures during the 2015/2016 El Nino (Bennett et al., 2021).
* * *
**Line 40**

*Reviewer Comment:*
 Is this really the case? I think you need to tone down this statement a bit as we don't have the complete evidence for this.

*Response:*
 We appreciate your caution. We have revised the statement to be more conservative:

Thus, field-based evidence suggests that African tropical forests might be especially resistant and resilient to climate extremes, but additional research is needed.
* * *
**Line 41-43**

*Reviewer Comment:*
 What is "Congolian"? Write Congo basin.

*Response:*
We appreciate your fine attention to detail. The Terrestrial Ecoregions of the World by Olson et al. (2001) define five different 'Congolian' forests and use 'Congolian' in their names, as can be seen in our first figure. Here we wrote 'Congolian tropical forest' as a descriptor to remain consistent with the official names for these ecoregions. Doing otherwise may confuse the reader as we do not show 'Congo Basin' on our map.
* * *
**Line 51**

*Reviewer Comment:*
 What about mortality? Was this not observed?

*Response:*
Thank you for your insightful comment regarding mortality. You raise an important point about its role in forest carbon dynamics. Mortality primarily affects the carbon stock, which is the total amount of carbon stored in forest biomass at a given time. When trees die, the carbon they contain is eventually released back into the atmosphere through decomposition, impacting the carbon stock.

In contrast, our discussion and our study focuses on changes in the carbon sink, which refers to the net flux of carbon between the forest ecosystem and the atmosphere resulting from ongoing processes like photosynthesis (carbon uptake) and respiration (carbon release). The net carbon sink is sensitive to short-term physiological responses of trees to environmental factors and climate anomalies.

We hope this clarifies the distinction between carbon stock and carbon sink and explains why our analysis focuses on the latter in relation to climate events.
* * *
**Line 53**

*Reviewer Comment:*
*What do you mean with "coastal forests"?*

*Response:*
We apologize for the ambiguity. We have clarified this as:

Also, these previous field-based analyses aggregated measurements annually at the continental scale although the field sampling was more commonly conducted in coastal forests, which are forests located closer to the coastlines that may have different environmental conditions and forest characteristics than interior forests.
* * *
**Line 56**

*Reviewer Comment:*
SIF is, I think, an indirect observation and hence still a proxy for productivity. Can you elaborate here.

*Response:*
Certainly. We have expanded the explanation by adding this sentence to the paragraph:

Because SIF is emitted during the light reactions of photosynthesis, it is directly sensitive to both the quantity of light absorbed and the efficiency with which that light is used for carbon fixation. This makes SIF a more direct proxy of photosynthetic activity and plant productivity compared to traditional vegetation indices, which primarily capture canopy greenness and structure.
* * *
**Line 59**

*Reviewer Comment:*
Leaf physiology: can you be more specific.

*Response:*
Sure, we have expanded the sentence to be more specific on what leaf physiology is:

SIF is a small amount of energy that is re-emitted by chlorophyll (1%-2%) and is sensitive to leaf physiology, which are the functions and processes within a plant leaf, including how it absorbs sunlight, exchanges gases through stomata, transports water and nutrients, and carries out photosynthesis (Johnson and Berry, 2021; Porcar-Castell et al., 2021, 2014).
* * *
**Line 66**

*Reviewer Comment:*
 VPD and temperature are linked; please make clear in the text.

*Response:*
 Sure, we have revised the sentence to clarify this relationship:

The studies that have utilized spaceborne SIF to investigate tropical Africa have found that (1) temperature and vapor pressure deficit (VPD)—which are interlinked because higher temperatures increase VPD by raising the air's capacity to hold moisture— control the productivity of African tropical forests…
* * *
**Line 82**

*Reviewer Comment:*
 Can you reformulate this hypothesis as not all forests in Africa you test are "moist", see Table S1.

*Response:*
 Good catch, thanks. We have reformulated it and clarified the statement to tie it into the paragraph that preceded it, which discussed what we know of the Amazon tropical forest.

Thus, we suspected that leaf demography and physiology in African tropical forests might respond to changes in environmental conditions in a manner similar to those observed in the Amazon.
* * *
**Line 107**

*Reviewer Comment:*
 Is 735-758 correct, i.e., it is not 743-758 nm?

*Response:*
It is correct, as they provide retrievals from two different windows for TROPOMI. See the abstract from Guanter et al. (2021):
https://essd.copernicus.org/articles/13/5423/2021/essd-13-5423-2021.pdf

Here is an excerpt:

*"Baseline SIF retrievals are derived from the 743–758 nm window. A secondary SIF dataset derived from an extended fitting window (735–758 nm window) is included."*
* * *
**Line 111**

*Reviewer Comment:*
*In the Results you also present soil moisture data. It is not clear how this is calculated from the Materials and Methods section.*

*Response:*
We apologize for the omission. We have added a description in the Methods section:

We used monthly averaged data from the ERA5-Land product (Muñoz Sabater, 2019), which is available in a spatial resolution of 0.1 degrees, for air temperature, photosynthetically active radiation (PAR) at the top of the canopy (PARTOC), VPD, and volumetric soil moisture (layer 1; 0-7 cm).
* * *
**Line 127**

*Reviewer Comment:*
*You indicate 4 vegetation indices and their pros and cons. But in the Results you focus on EVI and a little bit on NDVI. Please explain/justify better your choice.*

*Response:*
We have responded to this comment in the Main Comments section. Please see above.
* * *
**Line 147**

*Reviewer Comment:*
*Why this selection of African forest types and not the one proposed by Réjou-Méchain et al. 2021, Nature?*

*Response:*
The map shown by Réjou-Méchain et al. (2021) delineate forests according to functional type. Here are the two main reasons we used the Olson et al. (2001) ecoregion map.

1. Unfortunately, large swaths of forests are unclassified (marked as not calibrated). Excluding large expanses of forests within an ecoregion or plant functional type could have potentially biased our results. Excluding the data would have dramatically reduced the number of grid cells used in our statistical analyses, and would have introduced additional uncertainty.
2. The maps by Olson et al. (2001) have been used in about 10,000 studies, and are widely accepted. Olson et al. (2001) delineated ecoregions by considering distinct species assemblages, environmental conditions, and vegetation types. They characterized ecoregions based on unique combinations of flora and fauna, using

species distribution data—including the presence of endemic and specialist species—to define boundaries. Factors such as climate patterns, soil types, altitude, and hydrology were evaluated to reflect the ecological conditions influencing the distribution of species and ecosystems. Additionally, dominant vegetation forms (e.g., rainforest, savanna, mangroves) were used as indicators of ecological boundaries, as they reflect underlying environmental gradients and biological processes.

3. The Réjou-Méchain map unfortunately excludes two important ecoregions in West Africa, the Western and Eastern Guinean forests, which were an important scope of our study.
* * *
**Line 152-154**

*Reviewer Comment:*
*Where do we see this? Some of the text here is also redundant.*

*Response:*
This paragraph was under the 3 Results header, but it more aptly describes our classification method. Thus, we moved it to a new Methods section 2.8 and pointed the reader to the figures where the differences in precipitation variability and totals can be easily seen. This paragraph now reads:

2.8 West African and Central African Tropical forest

We noticed that the wettest ecoregions (Fig. 1) also had the highest variability in monthly total rainfall, and that there was a dissimilarity in our results among the wettest ecoregions with a high variability in monthly precipitation and the drier ecoregions with low variability (see precipitation bars in Figs. 2-3). Thus, we classified the 11 ecoregions into three groups according to their precipitation regime, monthly variability, and mean annual rainfall (Table S1). Four ecoregions in West Africa were characterized by seasonalities in mean monthly precipitation that had distinctive single wet and dry periods each year (Figs. 2-3), high monthly variability (sd ≥ 120 mm), and relatively high mean annual rainfall (> 2400 mm). We classified these ecoregions as West African moist tropical forest, which included the Cameroonian Highlands, Cross-Sanaga-Bioko Coastal Forest, Nigerian Lowlands and Niger Delta, and Western Guinean Lowlands. The six Central African ecoregions were characterized by seasonalities that typically had a double-peak pattern, low monthly variability (sd ≤ 100 mm), and relatively lower mean annual rainfall (< 2200 mm). We classified these forests as Central African tropical forests. The precipitation regime of the Eastern Guinean ecoregion in West Africa, which we classified as West African tropical forest, had mean annual rainfall (1544 mm) and monthly rainfall variability (81 mm) that was more similar to the Central African ecoregions.
* * *
**Line 155**

*Reviewer Comment:*
*I thought the classification into 11 regions was done a priori and not because of the seasonality.*

*Response:*
Right, the 11 ecoregions come from Olson et al. (2001). We further grouped these 11 ecoregions into 3 total regions for the sake of discussion. These three regions shared important characteristics, as we described in this paragraph. See above.
* * *
**Line 157**

*Reviewer Comment:*
*Elaborate better how seasonality can be seen, i.e., by better referring to Figure 1. But the legend and caption in Figure 1 is not clear and needs to be improved.*

*Response:*
Actually, we are referring to Figs. 2-3 here. We apologize. The sentence has now been revised as:

Four ecoregions in West Africa were characterized by seasonalities in mean monthly precipitation that had distinctive single wet and dry periods each year (Figs. 2-3), high monthly variability (sd ≥ 120 mm), and relatively high mean annual rainfall (> 2400 mm).
* * *
**Line 170**

*Reviewer Comment:*
*In this section moments of the year are given. However, I think this should be done more precisely by giving months of the year and not expressions like "mid-year", etc.*

*Response:*
The goal of this section is to provide a general description of the seasonality of SIF, environmental factors, and VIs. The peaks and minimums do not always occur in the same month in each ecoregion, and certainly not across all ecoregions. Parsing out the minimums and maximums for each year and for each ecoregion would detract from our goal to provide a general description of the patterns that are shown in our figures.
* * *
**Figure 2**

*Reviewer Comment:*

*The caption is not complete as also VPD is shown here. Justify why only "TROPOMI SIF" is shown (also valid for Figure 3).*

*Response:*

We apologize. VPD is shown in the caption, along with its unit. Perhaps the version of the PDF you received had a rendering error. We show only SIF on Figure 2 because including the vegetation indices would have rendered the figure incomprehensible, given that there would have been 7 lines in addition to the bars. For this reason, we show the vegetation indices in Figure 3. Both figures are below, as they are in the manuscript. Thank you.

[Figure]

**Fig. 2. Environmental conditions and solar-induced chlorophyll fluorescence for 11 African tropical forest ecoregions. Photosynthetically active radiation (PAR) is the amount of PAR at the top of the canopy (PAR_{TOC}). West African ecoregions are outlined in red.**

[Figure]

**Figure 3.** Monthly mean NDVI, EVI, SIF, and precipitation for 11 tropical forest ecoregions of Africa for 2019 - 2021. The shaded region delineates the year 2020. NDVI, EVI, and SIF share the left y-axis. West African ecoregions are outlined in red.
* * *
**Line 194**

*Reviewer Comment:*
*In this section a mostly qualitative description is given of synchrony with precipitation, PAR and VPD. I think more efforts can be done to make these relationships more quantitative. Hence my suggestion on top for zero-order and partial correlations,… And I am sure other techniques exist.*

*Response:*
Indeed, the goal of this section (3.2.1) was to provide a general description and interpretation of the seasonality of SIF, environmental factors, and VIs. The subsequent section (3.2.2) discusses the quantitative results of our zero-order correlation analyses (Pearson's and Spearman's). It is likely the reviewer's comment was made prior to reading section 3.2.2.

**Figure 4**

*Reviewer Comment:*
*I do not understand why a difference in correlations is proposed here; why VPD + Precip is given, etc.*

*Response:*

The goal here was to determine if SIF, EVI, or NDVI were increasingly related to VPD and less related to precipitation in forests with higher annual total rainfall and higher variability in monthly precipitation. We apologize that this was not clear. Please see the new paragraph referencing this figure and the revised caption:

To determine whether the correlations between SIF, EVI, or NDVI and VPD strengthened—and whether the correlation between SIF and precipitation weakened—as mean annual precipitation and the variability of monthly total precipitation increased, we compared the differences in correlation coefficients for VPD and precipitation across all sites. We found that, regardless of whether Pearson's or Spearman's correlation was used, the correlation between SIF and VPD strengthened while the correlation between SIF and precipitation weakened with increasing mean annual precipitation and greater variability in monthly precipitation (Fig. 4). This indicates that in forests with higher annual rainfall and more variable monthly precipitation, SIF becomes increasingly related to VPD and less related to precipitation. Conversely, NDVI showed a stronger correlation with precipitation and a weaker correlation with VPD in these same forests. However, this relationship is likely due to a saturated NDVI signal that mirrors the seasonality of precipitation in the West African moist tropical forest, as no significant correlation was found for EVI.

The caption for figure 4 had an typo, it is VPD plus precipitation. See the correction below, which also now explains r for VPD was always negative, and always positive for precip.

Figure 4. Regressions of the differences in the correlation coefficients for SIF, EVI, and NDVI vs VPD and precipitation using Pearson's and Spearman's correlation tests for each ecoregion. Differences are $r$ for VPD plus $r$ for precipitation (note that $r$ for VPD is always negative and $r$ for precipitation always positive). Top two rows are differences in Pearson's correlation coefficient, and bottom two rows are differences in Spearman's correlation coefficient.

**Figure 5**

*Reviewer Comment:*
*Can you explain why these periods are chosen?*

*Response:*
We have added the following to the caption of Figure 5:

The periods chosen correspond to the double-peak seasonality of precipitation for these tropical forests: January to April (increasing precipitation), April to July (decreasing precipitation), July to October (increasing precipitation), and October to January (decreasing precipitation).
* * *
**Figure 6**

*Reviewer Comment:*
The caption of this figure needs to be improved to read the figure independently.

*Response:*

Thank you for pointing out that this figure is lacking a better description. It has been revised as:

Fig. 6. Month in which minimum and maximum SIF, EVI, precipitation, volumetric soil water, and VPD occur in tropical African forests. For columns 2 and 4, the x-axis is month and the y-axis is the number of grid cells.
* * *
**Line 269**

*Reviewer Comment:*
It is surprising to see that the discussion is now mostly geared towards a comparison with the Amazon. Is this really the scope of the paper? And then suddenly the discussion is on PAR and VPD, while this was not emphasized in the introduction of the paper or the results section.

*Response:*
Yes, here our goal is to discuss the larger-scale implications of our findings in the context of the global tropical carbon cycle, of which are overwhelmingly composed of the Amazonian and African tropical forests. Note that the lead author of this manuscript has published extensively on SIF and environmental drivers in the Amazon, and so it is especially advantageous to describe our findings in the larger context of what is known of Earth's largest tropical rainforest.

We retitled this section and added a leading statement:

**4.1 Contrasting SIF dynamics and precipitation regimes in West African and Amazonian moist tropical forests**

Here, we situate our findings within the broader context of what is already known about other moist tropical forests—particularly those in the Amazon—to discern universal patterns and region-specific dynamics. This comparative perspective enhances our comprehension of how different moist tropical forests respond to variations in precipitation, vapor pressure deficit

(VPD), and photosynthetically active radiation (PAR), thereby contributing to more accurate models of global tropical forest behavior under changing environmental conditions.
* * *
**Line 305**

*Reviewer Comment:*
*But where are the physiology and phenology data? You refer here to the vegetation indices? This is not clear.*

*Response:*
Here we are referring to the spaceborne data, SIF (physiology) and EVI (phenology). We have clarified this statement:

"For instance, spaceborn-we observed that the physiology (SIF) and phenology (EVI) of the entire Central African tropical forest region acts in concert…"

---

## Author Comment (AC2)

**Dear Reviewer,**

We sincerely thank you for your thorough review of our manuscript and for your valuable comments and suggestions. Your insights have been instrumental in improving the clarity, depth, and overall quality of our work. Below, we address each of your comments in detail.
* * *
**Major Comments:**

**(1) Threshold Values for Ecoregion Grouping:**

*Comment:* The authors grouped the 11 ecoregions into three categories based on their precipitation regimes, using various thresholds such as high monthly variability (standard deviation ≥ 120 mm), relatively high mean annual rainfall (> 2400 mm), low monthly variability (standard deviation ≤ 100 mm), and relatively lower mean annual rainfall (< 2200 mm). However, the selection of these threshold values appears arbitrary, with no references provided to justify their choice.

*Response:*

We understand your concern about the selection of thresholds appearing arbitrary. However, considerable thought and analysis went into determining these groupings. Our classification was based on a combination of geographical location, environmental thresholds, and the observed characteristics of the seasonality in the spaceborne vegetation indices (VIs) and solar-induced chlorophyll fluorescence (SIF) data.

To address your comment and enhance clarity, we have moved and expanded upon our rationale, which was at the beginning of the results section, to the end of the methods in Section 2.8.:

2.8 West African and Central African Tropical Forests

We noticed that the wettest ecoregions also had the highest variability in monthly total rainfall, and that there was a dissimilarity in our results among the wettest ecoregions with high variability in monthly precipitation and the drier ecoregions with low variability. Thus, we classified the 11 ecoregions into three groups according to their precipitation regime, monthly variability, and mean annual rainfall (Table S1). Four ecoregions in West Africa were characterized by seasonalities that had distinctive single wet and dry periods each year, high monthly variability (standard deviation ≥ 120 mm), and relatively high mean annual rainfall (> 2400 mm). We classified these ecoregions as West African moist tropical forests, which included the Cameroonian Highlands, Cross-Sanaga-Bioko Coastal Forest, Nigerian Lowlands and Niger Delta, and Western Guinean Lowlands. The six Central African ecoregions were characterized by seasonalities that typically had a double-peak pattern, low monthly variability (standard

deviation ≤ 100 mm), and relatively lower mean annual rainfall (< 2200 mm). We classified these forests as Central African tropical forests. The precipitation regime of the Eastern Guinean ecoregion in West Africa had mean annual rainfall (1544 mm) and monthly rainfall variability (81 mm) that was more similar to the Central African ecoregions.

Additionally, we have included a new figure (Figure S1; below) in the supplementary materials. This figure plots the mean annual total precipitation and the standard deviation of monthly total precipitation for the 11 African tropical evergreen broadleaf ecoregions, color-coded according to the forest groups used in our study. This visual representation demonstrates the natural clustering of ecoregions based on their precipitation characteristics and supports our grouping methodology.

Our primary goal in grouping the ecoregions was to facilitate a meaningful discussion of regions with similar environmental conditions and observed seasonal patterns in SIF and VI data. The thresholds we selected emerged from our analysis of the data and were instrumental in highlighting the distinctions and similarities among the ecoregions. While these specific thresholds may not have been previously cited in the literature, they are grounded in the observed data and are appropriate for the context of our study.

Grouping ecoregions in this manner is a common practice when analyzing ecological data, as it allows for more nuanced interpretations and discussions of regional patterns and processes. Our approach does not necessitate prior usage in other research but is justified by the logical grouping based on observed environmental and phenological characteristics relevant to our study objectives.

We hope this explanation clarifies our rationale and assures you of the thoroughness of our methodology. Thank you again for your valuable feedback, which has helped us improve the clarity of our manuscript.

[Figure]

Figure S1. Mean annual total precipitation and standard deviation of monthly total precipitation in 2019-2021 for 11 African tropical evergreen broadleaf ecoregions, color coded according to the forest groups we used in our study.
* * *
**(2) Addressing the Second Hypothesis and Physiological Mechanisms:**

*Comment:* For the second hypothesis, the authors sought to test whether SIF would be more strongly linked to precipitation in less moist African forests and whether SIF and VPD would be positively correlated in moist forests. However, this hypothesis was not addressed in the Results and Discussion. There is a lack of explanation from the perspective of plant physiology regarding how these two distinct mechanisms—soil moisture deficit and atmospheric dryness—operate in African forests.

*Response:* We appreciate your pointing out that our second hypothesis was not adequately addressed. We have re-written section 3.2.2, now labeled section 3.3, to more clearly address our hypothesis and provide a deeper understanding as requested in major comment number 5. Here is the text of the new session.

Section 3.3

[revised manuscript text omitted]

These differing patterns suggest a potential decoupling between photosynthesis (as indicated by SIF) and canopy structure (as indicated by VIs like EVI and NIRv) in African tropical forests. In moist forests with high mean annual precipitation and variability, the canopy structure remains relatively constant throughout the year due to the evergreen nature of the forests. This results in stable EVI and NIRv values that are less sensitive to short-term environmental fluctuations. In contrast, photosynthetic activity, as indicated by SIF, can vary significantly in response to atmospheric conditions such as VPD.

One possible explanation for this decoupling is that SIF is more directly linked to the physiological status of leaves, capturing changes in photosynthetic efficiency and electron transport rates not necessarily reflected in canopy structural metrics. High VPD in moist forests can lead to stomatal closure to prevent excessive water loss, reducing $CO_2$ uptake and photosynthesis without significantly altering canopy structure or leaf area index (LAI). As a result, SIF decreases while EVI and NIRv remain relatively unchanged.

In less moist forests, where soil moisture deficits are more common, both photosynthesis and canopy structure can be affected by changes in precipitation. Limited water availability can lead to reduced leaf area through leaf shedding or inhibited growth, which is reflected in decreases in both SIF and VIs. This explains the stronger coupling between SIF and VIs in response to precipitation in these regions.

**(3) Enhancing the Introduction with African Forest Background:**

*Comment:* The Introduction should be reworked to emphasize the importance and current understanding of African forests. Now there is little description of the background of African forests. I would like to see (i) what is the climate background, (ii) what are the dominant vegetation species, and (iii) what is the current understanding of trends, inter-annual variations, and seasonality (most important since this is the topic of the study).

*Response:* Great suggestion! We have added two paragraphs to the introduction that follow the first. Here they are:

Rainfall seasonality in the tropical regions of West and Central Africa are primarily driven by the monsoon and the Intertropical Convergence Zone (ITCZ) (Longandjo and Rouault 2024). These two large-scale atmospheric processes create seasonal variations that bring distinct wet and dry seasons. Most areas near the equator including parts of Coastal West Africa and Central Africa experience two distinct passes of the ITCZ each year, resulting in a bimodal rainy season (Nicholson and Grist 2023). Byrne et al. (2018) reported a narrowing and strengthening of rainfall in the ITCZ over recent decades, based on satellite observations and simulation studies. However, their study found no evidence of a shift in the ITCZ's location.

The floristic composition and distribution of the tropical forests in the Guineo-Congolian region of West and Central Africa remain poorly sampled and understood (Sosef et al., 2017). Despite this, there is a general consensus that rainfall significantly influences floristic patterns across the region (Fayolle, 2014). Various authors have classified African tropical forests in different ways over time, including White (1979). However, a recent study by Fayolle et al. (2014) categorizes these forests into four groups: Wet-moist West Africa, Dry West Africa, Wet Central Africa, and Moist Central Africa. Using the RAINBIO database of tropical African vascular plant species, Sosef et al. (2017) reported a total of 22,577 species in the region. However, the authors emphasized that tropical forest biodiversity is still inadequately sampled.

**(4) Introducing Satellite Vegetation Indices Earlier:**

*Comment:* In the Introduction, the author mentions the use of satellite SIF to explore the relationship between photosynthesis and environmental factors. However, since satellite vegetation indices are also used later in the study, it would be beneficial to introduce these indices earlier and provide justification for the use of multiple satellite vegetation indicators.

*Response:* We agree with this suggestion, and Reviewer 1 had a similar comment. We now introduce VIs in the title and abstract. We also further elaborate on our focus of EVI over NDVI. Please see our responses to Reviewer 1.
* * *
**(5) Exploring Differing Patterns of SIF and Vegetation Indices:**

*Comment:* The differing patterns of SIF in comparison to EVI and NIRv in response to precipitation in Fig. 4 are interesting. I would expect the authors to explore this more thoroughly, providing a deeper explanation rather than only describing the empirical relationship. Does this suggest a decoupling between photosynthesis and canopy structure? Additionally, I would suggest the authors test whether the sampling times for SIF and MODIS vegetation indices are approximately the same. The authors mention that SIF retrieval is less sensitive to cloud cover, whereas MODIS vegetation indices are more affected by clouds.

*Response:* We have rewritten section 3.2.2 (now 3.3) to provide a deeper explanation; please see our response to major comment number 2 above.

It is not immediately clear what the implications would be for the differences in overpass time for MODIS and the satellites from which SIF is retrieved, as the reviewer seems to have something in mind here. SIF values are aggregated to daily values to account for differences in sampling time, as discussed in the methods section and the papers for the SIF datasets. Also, vegetation indices (or vegetation greenness) do not have diurnal patterns that would have affected comparisons. One of the reasons we chose a monthly time step at the ecoregion scale was to marginalize potential biases that might arise due to variances and differences in data acquisition from the different sensors.

We have added a section to the discussion that discusses the potential limitations of our study, which includes sensor characteristics and cloud cover. Please see our response below and Section 4.4.
* * *
**Minor Comments:**

**Abstract:**

**Line 16:** *The authors did not investigate actual photosynthesis; instead, they investigate SIF, which is an indicator of photosynthesis.*

*Response:* We agree! We have revised it as, "...to investigate the seasonality and synchrony of photosynthesis…".

**Line 17:** *Maybe delete the Amazon here since the aim of the study is not to compare African tropical forest and Amazon.*

*Response:* Thank you for this suggestion, but we see it important to compare and contrast our findings with what is known from field and spaceborne observations of the Amazon. Please see our response to Reviewer 1 on this point:

Our primary aim is to enhance the understanding of the seasonality of photosynthesis in African tropical forests and its synchrony with environmental factors. We chose to compare our findings with those from the Amazon for several important reasons:

*Benchmarking Against Well-Studied Systems*: The Amazon rainforest has been extensively studied, particularly regarding the seasonality of photosynthesis and its environmental drivers. By comparing African tropical forests to the Amazon, we can contextualize our results within a broader framework of tropical forest ecology. This comparison allows us to identify unique patterns and commonalities, thereby enriching the global understanding of tropical forest dynamics.

*Highlighting Regional Differences and Similarities*: Our comparison reveals both parallels and contrasts in the seasonality of photosynthesis between the two regions. For instance, while both regions exhibit increases in SIF during the dry season, the underlying environmental drivers and physiological responses differ. Discussing these differences enhances the scientific value of our work by highlighting how regional climatic conditions influence tropical forest productivity.

*Addressing Knowledge Gaps*: Given that African tropical forests are less studied compared to the Amazon, drawing parallels helps to fill knowledge gaps. It allows us to leverage the extensive body of research from the Amazon to interpret our findings and propose hypotheses about the mechanisms driving photosynthesis seasonality in Africa.

*Advancing Ecological Theory*: Comparing these two major tropical forest systems contributes to the development of general ecological theories about tropical forest functioning. It helps determine whether observed patterns are consistent across different continents or are region-specific due to unique environmental conditions.

*Informing Global Climate Models*: Understanding similarities and differences in photosynthetic responses is crucial for improving the accuracy of global carbon cycle models. By incorporating data from both African and Amazonian forests, we can better predict how tropical forests might respond to climate change on a global scale.

In light of these points, we believe that the comparison with the Amazon significantly enhances the interpretation and relevance of our findings. It not only aligns with the manuscript's aim but also provides a comprehensive perspective that benefits the broader scientific community.

Once again, we appreciate your feedback and hope this explanation clarifies the importance of including the comparison in our discussion.

**Introduction:**

**Line 25:** *"Carbon store" -> "carbon storage"?*

*Response:* Reviewer 1 suggested we change 'carbon store' to 'carbon stock', so we did so accordingly.

**Line 31-33:** *Add references.*

*Response:* We have added references.

**Line 35:** *What is the meaning of "carbon gains"? Carbon fluxes or storage?*

*Response:* We have reworded this to specify we are discussing carbon sink, "...assess gains and losses in the net carbon stock...".

**Paragraph Line 41:** *Please rephrase this paragraph. The first sentence mentions little work has been done to study the seasonality of West African tropical forests. However, the other part does not mention anything related to the seasonality. It seems the authors tried to introduce several knowledge gaps but without very clear and organized presentation.*

*Response:* Thank you, the paragraph now reads:

Satellite remote sensing studies have identified a double peak in the seasonality of leaf area and greenness in the Congolian tropical forests, which aligns with precipitation patterns. However, little research has been published on the seasonal dynamics of West African tropical forests, highlighting a significant knowledge gap. Understanding these seasonal patterns is crucial, especially amid debates over long-term vegetation changes in African tropical forests. While some studies have suggested a significant long-term browning trend in the Congolian forests, potentially linked to large-scale drying events (Zhou et al., 2014; Asefi-Najafabady and Saatchi, 2013; Jiang et al., 2019; Malhi and Wright, 2004), the most recent research found no widespread long-term decline in leaf area or greenness (Sun et al., 2022). This finding aligns with field observations showing no significant trend in the net carbon sink, suggesting that African tropical forests may be relatively resilient to climate variability. Addressing the lack of knowledge on the seasonality of West African tropical forests is therefore essential to fully understand the ecological dynamics and climate resilience of these ecosystems.

**Line 65:** *Maybe add Wang et al. (2023) in the introduction, which explores how different climate factors influence tropical forests using SIF.*

*Response:* Thank you for the suggestion, we have added Wang et al. (2023).

**Line 67-68:** *Not a complete sentence.*

*Response:* This sentence is a list that uses semicolons to separate the elements:

The studies that have utilized spaceborne SIF to investigate tropical Africa have found that (1) temperature and vapor pressure deficit (VPD)—which are interlinked because higher temperatures increase VPD by raising the air's capacity to hold moisture— control the productivity of African tropical forests (Madani et al., 2017; Umuhoza et al., 2023); (2) SIF tracks well the seasonality of photosynthesis, or gross primary productivity (GPP), over Africa (Mengistu et al., 2021); and (3) SIF has weak to insignificant relationships with VIs and VI-based APARchl (Doughty et al., 2021b).

**Line 75:** *Define ecoregions.*

*Response:* We have added the following statement to the beginning of Section 2.7 Ecoregions in the methods section:

An ecoregion is a substantial geographic area characterized by a unique composition of natural communities and ecosystems, where the majority of its species, ecological interactions, and environmental conditions are distinct and unified. These regions reflect the historical distribution of specific species and ecosystems and are categorized within broader biomes like forests, grasslands, or deserts, encompassing the diversity of terrestrial life on Earth.
* * *
**Methods:**

**Sections 2.1, 2.2, and 2.5:** *Not sure how the authors gridded these data to match climate reanalysis.*

*Response:* Actually, all data was used as-is. We did not grid the ungridded data and we did not re-grid any of the data. We have clarified this by moving Section 2.6 Copernicus Forest Cover to the beginning of the methods section and clarifying our approach:

We used both gridded and ungridded data in our analyses and these datasets are explicitly described below. All data were filtered using the 100-m Copernicus Land Cover dataset for the year 2019 (data after 2019 is not available) (Buchhorn et al., 2020), thus we only used data that fell within the forested areas. All data was used as-is without gridding or re-gridding, and values were aggregated to monthly timesteps at the ecoregion scale. To help ensure that our spaceborne data were acquired over forest and to reduce the likelihood of mixed pixels and soundings with mixed land cover types, we converted the forest land cover raster data to polygon and created a 2.5 km inner buffer.

**Section 2.4 ERA5 Reanalysis:** *I am not sure whether the author used CHIRPS Precipitation instead of ERA5 precipitation.*

*Response:* Thanks, we have clarified that we used CHIRPS precipitation data in what is now section 2.4 CHIRPS Precipitation:

Our precipitation data came from Climate Hazards group InfraRed Precipitation with Stations (CHIRPS), which is a long-term, near-global, daily data set.

**Section 2.7 Ecoregions:** *Please expand the description. It would be good to provide a map of ecoregions of the study area. How many types of Africa's tropical forest types do you have? How are they defined?*

*Response:* We have added a definition for Ecoregion, have elaborated on how ecoregions are defined, and now note how many forest ecogreions there are in Tropical Africa:

An ecoregion is a substantial geographic area characterized by a unique composition of natural communities and ecosystems, where the majority of its species, ecological interactions, and environmental conditions are distinct and unified. These regions reflect the historical distribution of specific species and ecosystems and are categorized within broader biomes like forests, grasslands, or deserts, encompassing the diversity of terrestrial life on Earth. We used the Terrestrial Ecoregions of the World boundaries (Olson et al., 2001) to distinguish between Africa's tropical forest types (Fig. 1), of which there are twelve. We combined the Nigerian Lowland Forests and the Niger Delta Swamp Forest ecoregions, which are adjacent to each other, in our analyses due to the sparsity of forest and spaceborne data for these forests.
* * *
**Results:**

**Lines 212, 215, and 216:** *"Sites" -> "ecoregions"?*

*Response:* Great catch, we have made these changes.

**Figure 4:** *Why don't you show the scattering plot at the pixel level?*

*Response:* As we now clarify in section 2.1 of the methods, all data was aggregated to monthly at the ecoregion scale. SIF data is not gridded, and is provided at the sounding level. There are no official gridded SIF datasets, as gridding SIF data is generally not advised. Variability and trends in SIF at a pixel level would be extremely biased by differences in spatial and temporal sampling, sun-sensor geometry, and sampling frequency. Also, uncertainty can be very high and variable at the pixel level for SIF data. I discuss these issues more fully in Section 5 of my paper on GOSAT, OCO, and OCO-2 SIF data (Doughty et al. 2021), and the regional-scale aggregation of SIF data is an approach that I have employed in nearly all of my papers.

**Section 3.3 Robustness of the SIF Signal:** *I think this part should go into Discussion or Supplementary. Also, there is no unit in Fig. 7.*

*Response:* We agree and have moved this section to the Discussion as section 4.4. There is not room on the y-axis for the label, so we included it in the legend. To further clarify, we have also added it to the figure caption.
* * *
**Discussion:**

**Line 271:** *Why suddenly discuss Amazon vs. African forests? I think the paper does not intend to compare Amazon and African forests since the authors did not mention any results of Amazon before.*

*Response:* Thank you for your opinion. We have responded to this concern in detail.

**Line 305:** *Not sure whether the authors have defined physiology and phenology.*

*Response:* We delineated this difference by adding the underlined text to our manuscript:

Recent advancements in the retrieval of solar-induced chlorophyll fluorescence (SIF) from space provides an observation-based method for monitoring plant physiology and the amount of PAR absorbed by chlorophyll ($APAR_{chl}$) and has been described as a proxy of photosynthesis (Doughty et al., 2019, 2021b). SIF is a small amount of energy that is re-emitted by chlorophyll (1%-2%) and is sensitive to leaf physiology, which are the functions and processes within a plant leaf, including how it absorbs sunlight, exchanges gases through stomata, transports water and nutrients, and carries out photosynthesis (Johnson and Berry, 2021; Porcar-Castell et al., 2021, 2014). Because SIF is emitted during the light reactions of photosynthesis, it is directly sensitive to both the quantity of light absorbed and the efficiency with which that light is used for carbon fixation. This makes SIF a more direct proxy of photosynthetic activity and plant productivity compared to traditional vegetation indices, which primarily capture canopy greenness and structure.
* * *
**Additional Suggestion:**

*The authors may add a paragraph to describe the uncertainty or limitation of this study.*

*Response:* Thank you for this important suggestion. We have added this section to the Discussion:

4.4 Uncertainties and Limitations

[revised manuscript text omitted]

---

## Author Response (AR2)

Dr. Yang,

Thank you very much for your careful review of our manuscript and our responses. We are more than happy to make the requested changes. Please see our responses below to your suggestions.

Sincerely,

Russell Doughty and coauthors
* * *
*1) The legend in Figure 1 is too small. I have to zoom in a lot to see the text. Consider moving it outside the main image area.*

*Response:*
We agree! To save space in the final publication and to facilitate a timely response, we have described the legend in the figure caption.
* * *
*2) Several sentences in the discussion need to be supported by either the current work or others' paper. For example, in Section 4.3., the authors started with "Our study encompassed several tropical forest ecoregions with mean annual precipitation at or below 1700 mm, such as the 399 Eastern Guinean, Western Congolian Swamp Forest, and Northwestern Congolian Lowland Forest. " which can be referred back to Figure 1.*

*Response:*
Thank you. We now cite Figure 1 here as requested.
* * *
*Moreover, the authors wrote "First, trees in drier evergreen forests may have evolved mechanisms to optimize water use efficiency, such as closing stomata during periods of high VPD to reduce transpiration losses, which in turn affects photosynthesis. Second, these forests might rely on deep root systems to access groundwater during dry periods, but prolonged low precipitation can still lead to water deficits affecting canopy function. Finally, the tree species in these regions may have traits that are adapted to lower water availability, such as being semi-deciduous, influencing the overall ecosystem response to environmental drivers. " These are all reasonable points, but they should be supported by properly citing other works. Otherwise, it may read as hand-waving to the readers.*

*Response:*

We now provide citations for each of these statements. Thank you very much for pointing this out; the changes improve our manuscript.
* * *
*3) I would reconsider the argument in Section 3.3. Here, the authors wrote, "In less moist forests (mean annual precipitation < 2000 mm), we found that SIF was significantly positively correlated with precipitation and significantly negatively correlated with VPD (Figs. S1 and S2). This suggests that soil moisture availability is a key driver of photosynthesis in these regions; limited precipitation leads to soil moisture deficits that constrain plant physiological processes." Although we can interpret this as that more soil moisture leads to more SIF emission, the other side of the result is high VPD leads to lower SIF. Thus the result does not support one or the other, but it indicates that both may have played a role. Reviewer 2 also had a similar comment.*

*Response:*
This is a great point! We have revised that argument as:

In less moist forests (mean annual precipitation < 2000 mm), SIF was significantly positively correlated with precipitation and significantly negatively correlated with VPD (Figs. S1 and S2). These dual relationships suggest that photosynthetic activity in these regions is driven by both soil moisture availability and atmospheric water demand. On one hand, limited precipitation reduces soil water content, constraining stomatal conductance and thereby lowering photosynthesis. On the other hand, high VPD can prompt stomatal closure to mitigate water loss, which in turn decreases carbon uptake and SIF. Hence, the interplay between precipitation and VPD jointly regulates photosynthetic fluxes, and both factors should be considered when interpreting SIF variations in less moist forests.